# ALDH7A1 inhibits the intracellular transport pathways during hypoxia and starvation to promote cellular energy homeostasis

Jia-Shu Yang[1], Jia-Wei Hsu[1], Seung-Yeol Park[1,2], Stella Y. Lee[3], Jian Li[1], Ming Bai[1], Claudia Alves [1], William Tseng[1], Xavier Michelet[1], I-Cheng Ho[1] & Victor W. Hsu[1]

The aldehyde dehydrogenase (ALDH) family of metabolic enzymes converts aldehydes to carboxylates. Here, we find that the reductive consequence of ALDH7A1 activity, which generates NADH (nicotinamide adenine dinucleotide, reduced form) from NAD, underlies how ALDH7A1 coordinates a broad inhibition of the intracellular transport pathways. Studying vesicle formation by the Coat Protein I (COPI) complex, we elucidate that NADH generated by ALDH7A1 targets Brefeldin-A ADP-Ribosylated Substrate (BARS) to inhibit COPI vesicle fission. Moreover, defining a physiologic role for the broad transport inhibition exerted by ALDH7A1, we find that it acts to reduce energy consumption during hypoxia and starvation to promote cellular energy homeostasis. These findings advance the understanding of intracellular transport by revealing how the coordination of multiple pathways can be achieved, and also defining circumstances when such coordination is needed, as well as uncovering an unexpected way that NADH acts in cellular energetics.

[1] Division of Rheumatology, Immunology and Allergy, Brigham and Women's Hospital, and Department of Medicine, Harvard Medical School, Boston, MA 02115, USA. [2] Department of Life Sciences, Pohang University of Science and Technology, Pohang, Gyeongbuk 37673, Republic of Korea. [3] Division of Biology, Kansas State University, Manhattan, KS 66506, USA. Correspondence and requests for materials should be addressed to J.-S.Y. (email: jyang@rics.bwh.harvard.edu) or to V.W.H. (email: vhsu@bwh.harvard.edu)

Much of our current understanding of vesicular transport comes from studies on model pathways[1–8]. One such pathway involves transport by the coat protein I (COPI) complex[9–11]. Early studies identified a multimeric complex, known as coatomer, as the core components of the COPI complex, and ADP-ribosylation factor 1 (ARF1) as the small GTPase that regulates the recruitment of coatomer from the cytosol to Golgi membrane to initiate COPI vesicle formation[12,13]. Subsequently, additional key factors have been identified to act in this process. These include the GTPase-activating protein (GAP) for ARF1, known as ARFGAP1, which acts as another component of the COPI complex[14,15], and Brefeldin-A ADP-ribosylated substrate (BARS), which acts in COPI vesicle fission[16,17], the stage of vesicle formation when coated buds undergo neck constriction to become released from the membrane as transport vesicles.

To gain further insight into how COPI vesicle formation is achieved, we have been seeking to identify additional factors that act in this process. In this study, we show that a member of the aldehyde dehydrogenase (ALDH) family of metabolic enzymes, ALDH7A1, also known as antiquitin[18], inhibits COPI vesicle fission by targeting BARS. As BARS acts in the fission stage of multiple transport pathways, we then uncover a broader set of intracellular pathways inhibited by ALDH7A1. We also elucidate a physiologic role for this global inhibition, showing that it promotes cellular energy homeostasis by reducing energy consumption during hypoxia and starvation, situations when cellular energy production is impaired.

## Results

### ALDH7A1 inhibits COPI vesicle fission by targeting BARS.

When BARS is fused to glutathione-S-transferase (GST), and then incubated with cytosol in a pulldown experiment, we identified multiple interacting proteins (Supplementary Table 1). Besides transport factors, we also identified metabolic enzymes. In this regard, we have recently uncovered an unconventional role for glyceraldehyde 3-phosphate dehydrogenase (GAPDH) in regulating COPI vesicle fission by targeting ARFGAP1[19]. Thus, we explored whether any of the metabolic enzymes implicated to interact with BARS could also have a role in COPI transport.

A quantitative assay that tracks COPI transport in cells has been established, which examines the redistribution of a COPI cargo protein (known as VSVG-KDELR) from the Golgi complex to the endoplasmic reticulum (ER)[16,17,20]. Performing this assay, we found that siRNA against *ALDH7A1* (Supplementary Fig. 1a), but not against *catalase* (Supplementary Fig. 1b) or *formimidoyltransferase-cyclodeminase* (Ftcd) (Supplementary Fig. 1c), enhances COPI transport (Supplementary Fig. 1d). We also confirmed that ALDH7A1 can interact directly with BARS, as assessed by a pulldown experiment using purified components (Fig. 1a). Moreover, ALDH7A1 interacts with BARS in cells, as assessed by a co-precipitation study (Fig. 1b).

We then examined whether the catalytic activity is needed for ALDH7A1 to affect COPI transport. We targeted a residue in the catalytic domain that is conserved across ALDH members[21], mutating glutamate at position 268 to glutamine (E268Q), and found that the point mutant could not rescue the effect of siRNA against *ALDH7A1* (Fig. 1c) on COPI transport (Fig. 1d and Supplementary Fig. 2a). Moreover, we found that ALDH7A1 overexpression had the opposite effect of inhibiting COPI transport, which was also prevented by the point mutation (Fig. 1e and Supplementary Fig. 2b). Thus, these results suggested that ALDH7A1 acts as a negative regulator of COPI transport.

To gain further insight into how ALDH7A1 inhibits COPI transport, we next pursued the reconstitution of COPI vesicles from Golgi membrane, as this approach enables the details of COPI vesicle formation to be dissected out[16,17,20]. When ALDH7A1 was added as another purified component (Supplementary Fig. 2c), we found that COPI vesicle formation is inhibited (Fig. 1f). We also found that the catalytic mutation prevents the inhibition of COPI vesicle formation in the reconstitution system (Fig. 1f). Examination by electron microscopy (EM) revealed that ALDH7A1 induces the accumulation of buds with constricted necks on Golgi membrane (Fig. 1g). Thus, these results suggested that ALDH7A1 acts as a negative regulator of COPI transport by inhibiting the fission stage of vesicle formation.

We next considered that ALDH7A1 activity has been characterized to convert multiple aldehydes to their corresponding carboxylates[22]. Examining these metabolic products, we found that none could inhibit COPI vesicle formation (Fig. 1h). We then noted that nicotinamide adenine dinucleotide in reduced form (NADH) has been shown previously to inhibit the function of BARS in COPI vesicle fission[16]. Thus, we explored whether NADH generated by ALDH7A1 activity could target BARS in explaining how ALDH7A1 inhibits COPI vesicle fission.

We first confirmed that simply adding NADH to the reconstitution system inhibits COPI vesicle formation (Fig. 1h). We also sought confirmation that NADH can directly target BARS. We had previously incubated recombinant BARS with liposomes generated from defined pure lipids to demonstrate that BARS can directly induce membrane deformation[17]. Re-visiting this assay, we found that the addition of NADH prevents BARS from inducing liposome tubulation (Supplementary Fig. 2d).

To further confirm that ALDH7A1 inhibits BARS through the generation of NADH, we next examined the effect of a point mutation in BARS, glycine at position 172 to glutamate (G172E), which has been shown previously to render BARS insensitive to NADH inhibition[16]. When wild-type BARS was replaced with this G172E mutant, we found that ALDH7A1 could no longer inhibit COPI vesicle formation (Fig. 1i). Moreover, the G172E mutation prevents the ability of NADH to inhibit liposome tubulation by BARS (Supplementary Fig. 2d). We also sought physiologic confirmation by pursuing cell-based studies. We replaced endogenous wild-type BARS with the mutant (G172E) BARS, which was achieved by treating cells with siRNA against *BARS* and then transfecting the mutant BARS, and found that ALDH7A1 overexpression could no longer inhibit COPI transport (Fig. 1j). Thus, multiple lines of evidence all pointed to ALDH7A1 activity inhibiting COPI vesicle fission through the generation of NADH that targets BARS.

### ALDH7A1 also inhibits multiple other intracellular pathways.

We next considered that BARS also acts at the fission stage in other transport pathways[23]. Thus, we explored whether ALDH7A1 regulates additional pathways besides COPI transport. We had previously pursued a quantitative screen of the major intracellular pathways, which tracks the transit of model cargoes in transport pathways through their colocalization with organelle markers in time-course studies[24]. Pursuing this approach, we found that siRNA against *ALDH7A1* does not affect transport from the ER to the Golgi (Fig. 2a and Supplementary Fig. 3a). In contrast, siRNA against *ALDH7A1* enhanced transport from the Golgi to the plasma membrane (PM) (Fig. 2b and Supplementary Fig. 3b).

All three major endocytic routes were also enhanced by siRNA against ALDH7A1, as reflected by the enhanced recycling of transferrin receptor (TfR) to the PM (Fig. 2c and Supplementary Fig. 3c), the enhanced retrograde transport of internalized cholera toxin (CT) to the Golgi (Fig. 2d and Supplementary Fig. 3d), and the enhanced delivery of internalized dextran to the lysosome

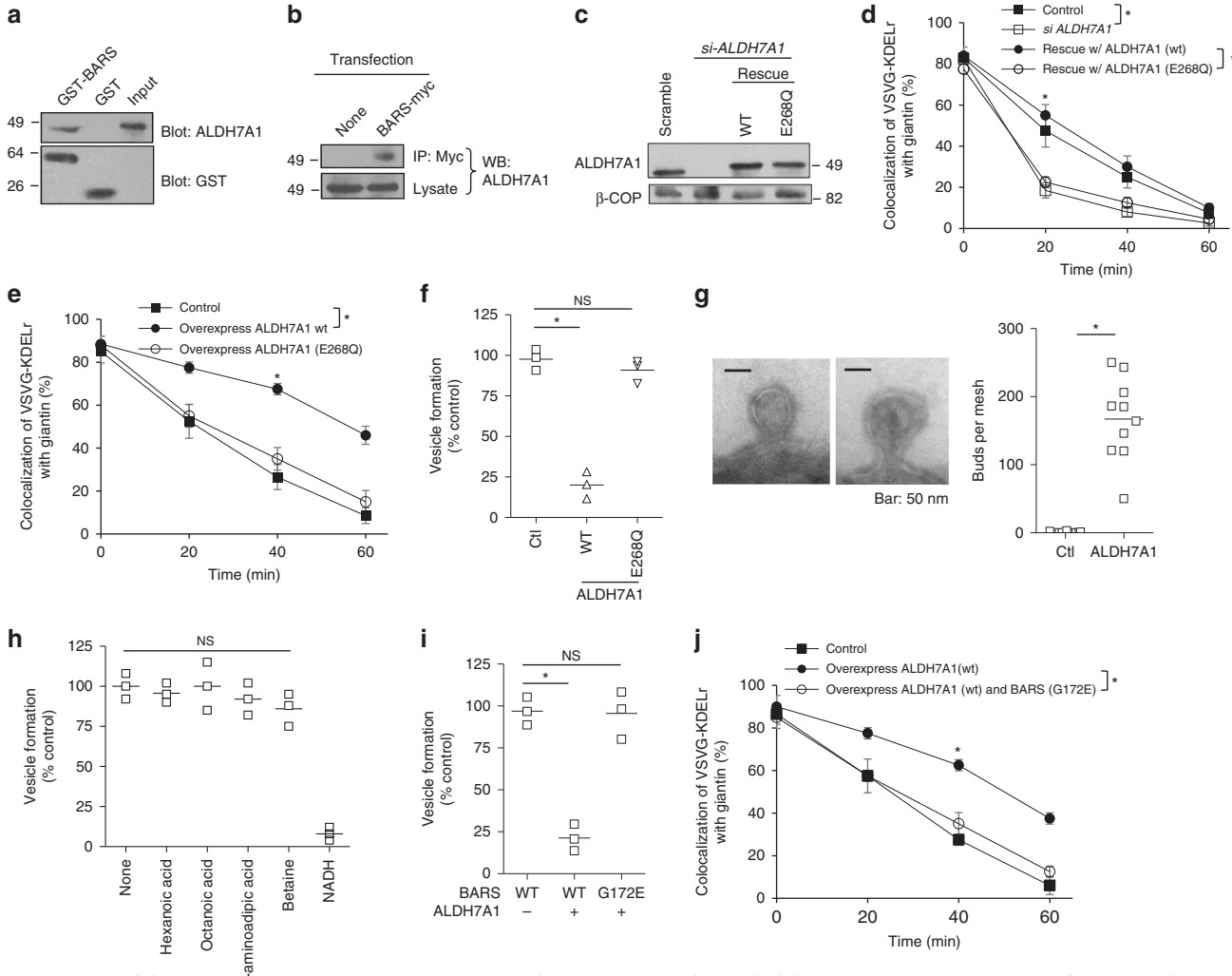

**Fig. 1** ALDH7A1 inhibits COPI transport. Quantitative results are shown as mean with standard deviation; *$p < 0.05$, NS not significant, paired two-tailed Student's *t*-test. Source data are provided as a Source Data file. **a** GST–BARS on beads was incubated with recombinant ALDH7A1, followed by immunoblotting with antibodies against indicated proteins; $n = 2$ independent experiments with a representative result shown. **b** HeLa cells were transfected with construct as indicated, followed by immunoprecipitation for the myc tag and then immunoblotting for ALDH7A1; $n = 2$ independent experiments with a representative result shown. **c** HeLa cells were treated as indicated and then whole-cell lysates were immunoblotted for proteins as indicated; $n = 2$ independent experiments with a representative result shown. Rescues for siRNA-treated cells were performed using tagged forms of ALDH7A1. **d** HeLa cells were treated as indicated, and then colocalization of VSVG-KDELR with a *cis*-Golgi marker (giantin) coupled with kinetic analysis was performed; $n = 3$ independent experiments. **e** HeLa cells were treated as indicated, and then colocalization of VSVG-KDELR with a *cis*-Golgi marker (giantin) coupled with kinetic analysis was performed; $n = 3$ independent experiments. **f** The COPI reconstitution system was performed followed by quantitation; $n = 3$ independent experiments. **g** Golgi membrane after the incubation in the reconstitution system was examined by EM. Representative images are shown on the left (scale bar, 50 nm), and quantitation is shown on the right; $n = 3$ independent experiments. **h** The COPI reconstitution system was performed in the presence of different metabolic products (carboxylates) of ALDH7A1 or NADH; NS not significant, ANOVA; $n = 3$ independent experiments. **i** The COPI reconstitution system was performed followed by quantitation; $n = 3$ independent experiments. **j** HeLa cells were treated as indicated, and then the colocalization of VSVG-KDELR with a *cis*-Golgi marker (giantin) coupled with kinetic analysis was performed; $n = 3$ independent experiments

(Fig. 2e and Supplementary Fig. 3e). Fluid-phase endocytosis was also enhanced, as reflected by increased uptake of dextran (Fig. 2f and Supplementary Fig. 3f). However, clathrin-mediated endocytosis was unaffected, as tracked by the internalization of TfR to the early endosome (Fig. 2g). Altogether, the results revealed that six out of the eight major intracellular pathways are targeted by ALDH7A1 for inhibition (summarized in Fig. 2h).

We also considered that the enhanced delivery of dextran to the lysosome induced by siRNA against *ALDH7A1* (see Fig. 2e) could be an indirect effect of fluid-phase endocytosis having been enhanced by the siRNA treatment (see Fig. 2f). Thus, we examined another endocytic cargo that is transported to the lysosome, but whose internalization at the PM is not affected by

siRNA against *ALDH7A1*. The epidermal growth factor (EGF) receptor (EGFR) is predicted to be one such cargo, as it is internalized through clathrin-mediated endocytosis, and then transported through the endosome to reach the lysosome[6]. We first confirmed that the endocytosis of EGFR is unaffected by siRNA against *ALDH7A1* (Fig. 2i). Subsequently, tracking the fate of internalized EGFR to lysosomes, we found that this transport is also enhanced by siRNA against *ALDH7A1* (Fig. 2j). Thus, these results further confirmed that ALDH7A1 inhibits endocytic transport to the lysosome.

We next found that the overexpression of ALDH7A1 has the opposite effect of the siRNA treatment in affecting the identical pathways (Fig. 3a–i). These effects also require the catalytic

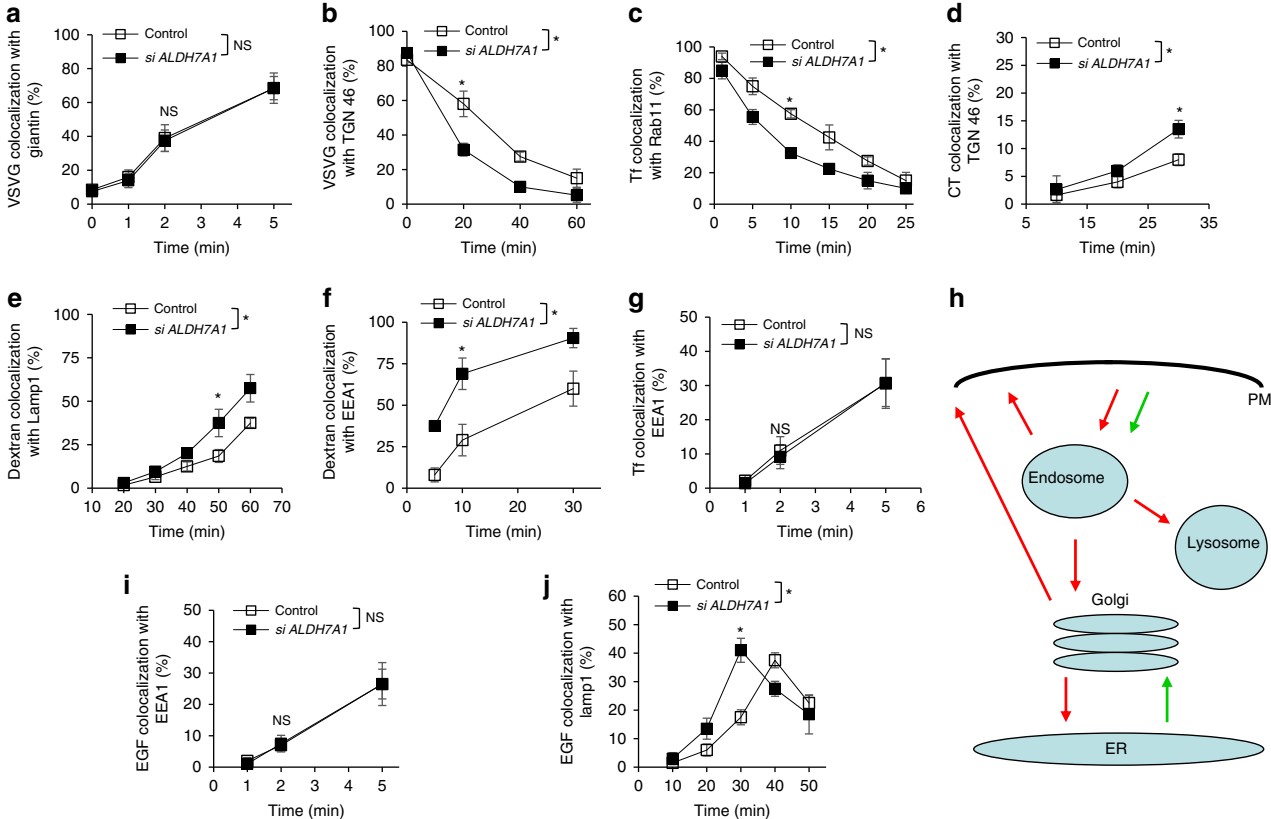

**Fig. 2** Knocking down *ALDH7A1* enhances intracellular pathways. Quantitative results are shown as mean with standard deviation; *$p < 0.05$, NS not significant, paired two-tailed Student's *t*-test; $n = 3$ independent experiments. Source data are provided as a Source Data file. HeLa cells were assessed for the following transport pathways: **a** Transport of VSVG from the ER to the Golgi was assessed through the quantitative colocalization of VSVG with a *cis*-Golgi marker (giantin). **b** Transport of VSVG from the Golgi to the plasma membrane was assessed through the quantitative colocalization of VSVG with a TGN marker (TGN46). **c** Endocytic recycling of Tf from the recycling endosome (RE) to the plasma membrane was assessed through the quantitative colocalization of internalized Tf with a recycling endosome marker (Rab11). **d** Endocytic transport of cholera toxin (CT) to the Golgi was assessed through the quantitative colocalization of internalized CT with a TGN marker (TGN46). **e** Endocytic transport of internalized dextran to the lysosome was assessed through the quantitative colocalization of internalized dextran with a lysosome marker (Lamp1). **f** Fluid-phase endocytosis of dextran was assessed through the quantitative colocalization of internalized dextran with an early endosome marker (EEA1). **g** Clathrin-mediated endocytosis of Tf was assessed through the quantitative colocalization of internalized transferrin (Tf) with an early endosome marker (EEA1). **h** Summary of the major intracellular pathways regulated by ALDH7A1. Red arrows indicate pathways inhibited by ALDH7A1. Green arrows indicate pathways unaffected by ALDH7A1. **i** Endocytosis of EGFR was assessed through the quantitative colocalization of internalized EGF with an early endosome marker (EEA1). **j** Endocytic transport of internalized EGFR to the lysosome was assessed through the quantitative colocalization of internalized EGF with a lysosome marker (Lamp1)

activity of ALDH7A1, as the catalytic mutation (E268Q) prevented the ability of ALDH7A1 overexpression from exerting transport inhibition (Fig. 3a–i). Moreover, ALDH7A1 over-expression could no longer inhibit the BARS-dependent pathways, when endogenous wild-type BARS was replaced with the mutant (G172E) BARS that is defective in binding to NADH (Supplementary Fig. 3g and h). Consistent with these findings, we also found that enhanced COPI transport induced by siRNA against *ALDH7A1* is prevented when cells were treated with siRNA against *BARS* (Supplementary Fig. 3i).

We also confirmed that these effects of ALDH7A1 are not restricted to HeLa cells. Examining human embryonic kidney 293 (HEK293) cells, we observed similar effects, as reducing ALDH7A1 level through siRNA treatment enhances transport in the identical pathways and increasing ALDH7A1 level through overexpression inhibits transport in the identical pathways (Supplementary Fig. 4a–i).

We next examined whether the affected transport pathways could all be explained by ALDH7A1 targeting BARS. Upon siRNA against *BARS*, we observed transport inhibition in Golgi-to-ER transport (Supplementary Fig. 5a), Golgi-to-PM transport

(Supplementary Fig. 5b), and fluid-phase endocytosis (Supplementary Fig. 5c). In contrast, this siRNA treatment did not affect endocytic recycling (Supplementary Fig. 5d), endocytic transport to the lysosome (Supplementary Fig. 5e), and endocytic transport to the Golgi (Supplementary Fig. 5f). Thus, ALDH7A1 is predicted to target additional factor(s) besides BARS in explaining how it exerts a broad inhibition of the transport pathways.

Next, to rule out that any metabolic enzyme that generates NADH could exert transport inhibition, we examined the effect of reducing the cellular level of two dehydrogenases, aldolase (Supplementary Fig. 5g) and malate dehydrogenase 2 (MDH2) (Supplementary Fig. 5h). However, transport was not affected by siRNA targeting against either enzyme (Supplementary Fig. 5i). We also examined another member of the ALDH family, and found that siRNA against *ALDH1A1* (Supplementary Fig. 6a) does not affect the transport pathways (Supplementary Fig. 6b–h). Moreover, consistent with how ALDH7A1 affects the BARS-dependent pathways, we found that ALDH1A1, which does not affect the BARS-dependent pathways, also does not interact with BARS (Supplementary Fig. 6i).

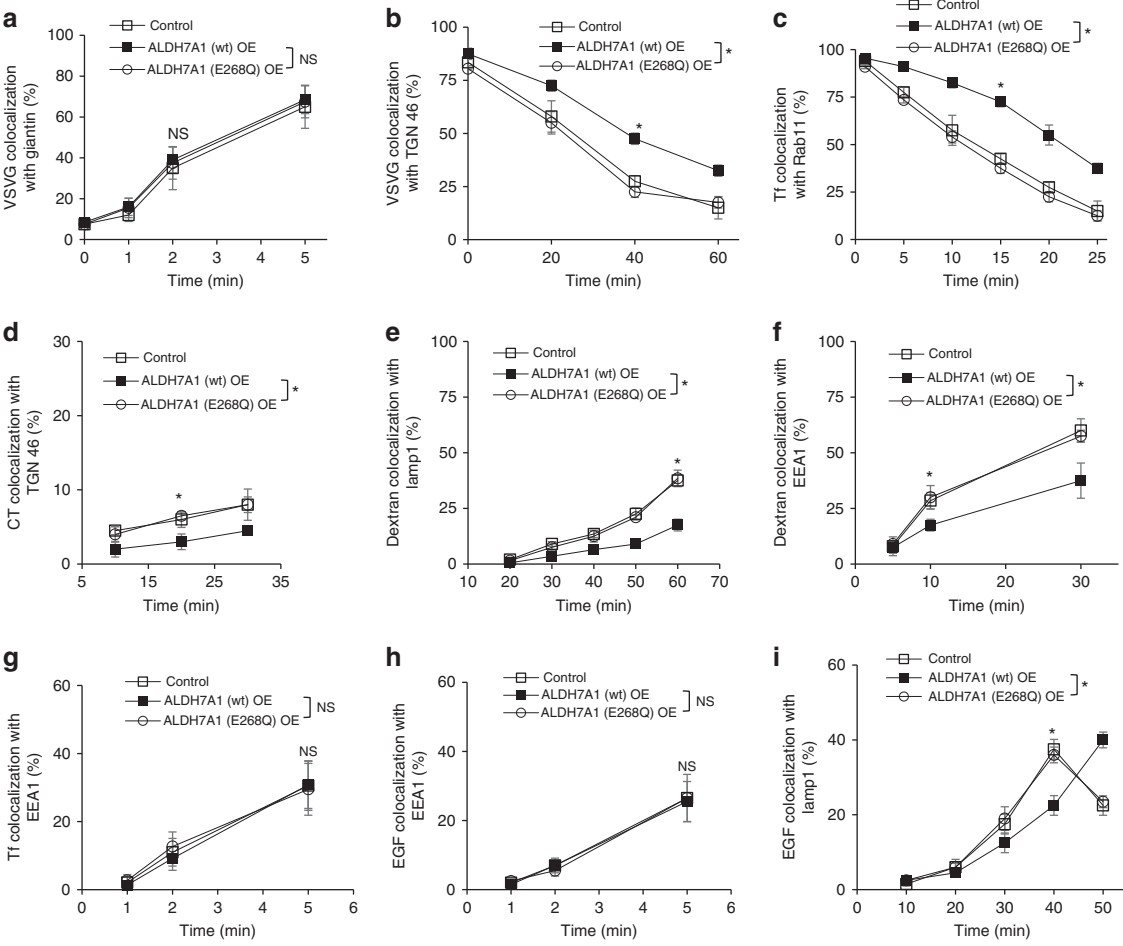

**Fig. 3** ALDH7A1 overexpression inhibits intracellular pathways. Quantitative results are shown as mean with standard deviation; *$p < 0.05$, NS not significant, paired two-tailed Student's $t$-test; $n = 3$ independent experiments. Source data are provided as a Source Data file. HeLa cells were assessed for the following transport pathways: **a** Transport of VSVG from the ER to the Golgi was assessed through the quantitative colocalization of VSVG with a *cis*-Golgi marker (giantin). **b** Transport of VSVG from the Golgi to the plasma membrane was assessed through the quantitative colocalization of VSVG with a TGN marker (TGN46). **c** Endocytic recycling of Tf from the recycling endosome (RE) to the plasma membrane was assessed through the quantitative colocalization of internalized Tf with a recycling endosome marker (Rab11). **d** Endocytic transport of CT to the Golgi was assessed through the quantitative colocalization of internalized CT with a TGN marker (TGN46). **e** Endocytic transport of internalized dextran to the lysosome was assessed through the quantitative colocalization of internalized dextran with a lysosome marker (Lamp1). **f** Fluid-phase endocytosis of dextran was assessed through the quantitative colocalization of internalized dextran with an early endosome marker (EEA1). **g** Clathrin-mediated endocytosis of Tf was assessed through the quantitative colocalization of internalized transferrin (Tf) with an early endosome marker (EEA1). **h** Endocytosis of EGFR was assessed through the quantitative colocalization of internalized EGF with an early endosome marker (EEA1). **i** Endocytic transport of internalized EGFR to the lysosome was assessed through the quantitative colocalization of internalized EGF with a lysosome marker (Lamp1)

To further characterize transport inhibition by ALDH7A1, we next fractionated the cell into total membrane and cytosol, and found that ALDH7A1 interacts with BARS on the membrane but not in the cytosol (Fig. 4a). Moreover, we found by confocal microscopy that ALDH7A1 is broadly distributed among the intracellular membrane compartments (Fig. 4b). Notably, ALDH7A1 level is markedly reduced at the ER (Fig. 4c), which is consistent with our observation above that ALDH7A1 does not affect transport from the ER.

We also considered that BARS acts not only as a fission factor in transport pathways, but also as a transcription repressor. In this latter role, it is known as C-terminal binding protein (CtBP)[25]. Thus, we examined whether the ALDH7A1 could affect the transport pathways indirectly through the transcription function of BARS. To act in transcription, BARS needs to localize to the nucleus. Thus, to prevent the nuclear localization of BARS, we appended a nuclear export signal (NES) to BARS (NES-BARS). We initially confirmed by immunofluorescence microscopy (Fig. 4d)

and subcellular fractionation (Fig. 4e) that appending NES to BARS markedly reduces its nuclear localization. Next, we replaced endogenous wild-type BARS with NES-BARS, which was achieved by knocking down BARS followed by rescue using a siRNA-resistant form of NES-BARS. We then confirmed functionally that NES-BARS has markedly reduced the ability to inhibit transcription, as assessed through the mRNA level of E-cadherin (Fig. 4f), which has been a model transcript for studies on transcriptional repression by CtBP[25]. Notably, we then found that NES-BARS mediates the inhibition of transport pathways by ALDH7A1, as reflected by effects on COPI transport (Fig. 4g) and fluid-phase uptake (Fig. 4h). Thus, the collective results suggested that transport inhibition by ALDH7A1 is unlikely to involve the targeting of the transcription function of BARS/CtBP.

**ALDH7A1 promotes cellular energy homeostasis.** We then sought to elucidate the physiologic significance of the broad transport inhibition exerted by ALDH7A1. In plants, ALDH7A1

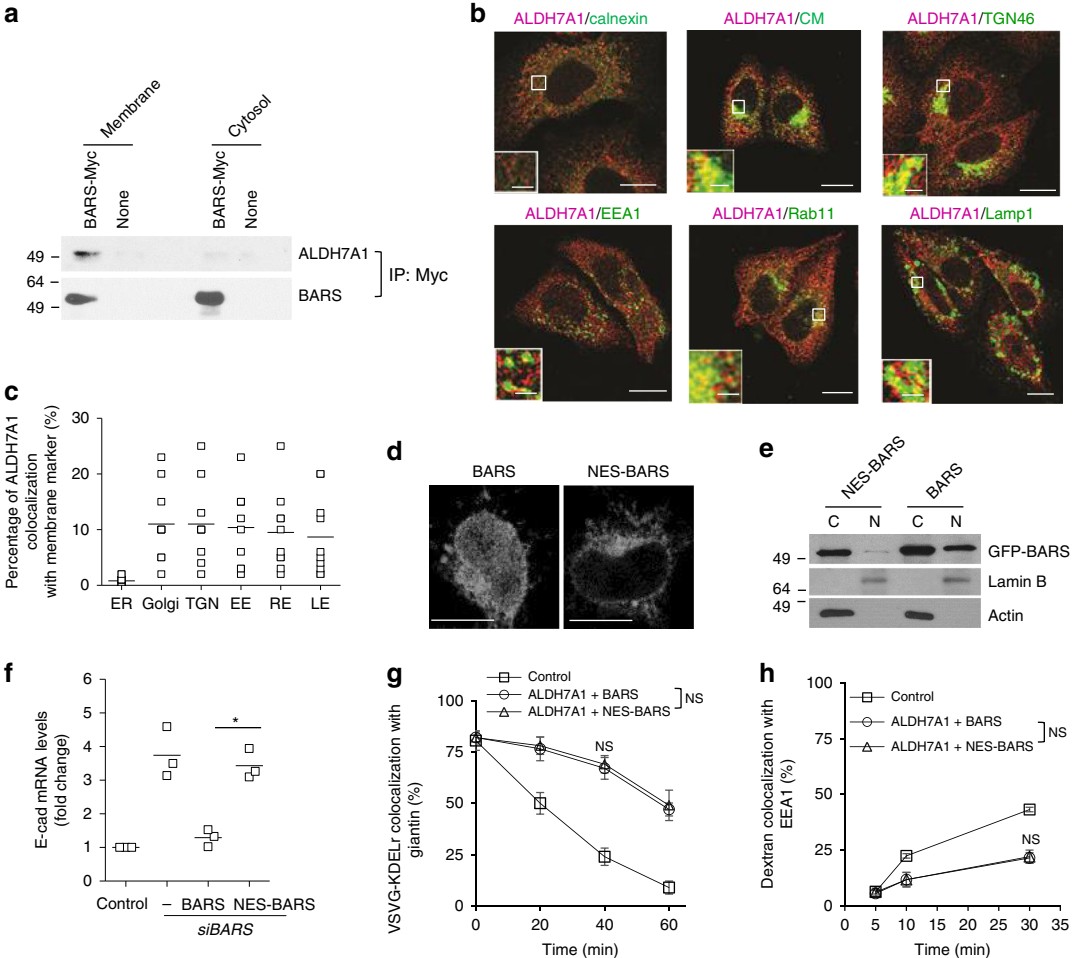

**Fig. 4** ALDH7A1 targets the transport function of BARS. Source data are provided as a Source Data file. **a** HeLa cells were subjected to subcellular fractionation to obtain total membrane vs. cytosol fractions. Both fractions were then subjected to co-immunoprecipitation experiments to detect the association of ALDH7A1 with BARS; $n = 2$ independent experiments with a representative result shown. **b** Confocal microscopy was performed to detect the colocalization of ALDH7A1 with different organelle markers; scale bar, 10 μm (2 μm in inset); $n = 3$ independent experiments with representative images shown. **c** Quantitation of the colocalization results above; $n = 3$ independent experiments with ten fields of cells examined in each experiment. **d** HeLa cells were transfected with GFP-tagged NES-BARS followed by confocal imaging across the nuclear portion of the cell; scale bar, 10 μm; $n = 2$ independent experiments with a representative image shown. **e** HeLa cells were fractionated into nuclear vs. cytoplasmic fractions, followed by immunoblotting for proteins as indicated. Lamin B serves as a marker of the nucleus, while actin serves as a marker of the cytoplasm; $n = 2$ independent experiments with a representative result shown. **f** HeLa cells were treated as indicated, and then the mRNA level of E-cadherin, which has been a model transcript for studies on transcription repression by BARS/CtBP, is quantified; $n = 3$ independent experiments. **g** HeLa cells were treated as indicated, and then colocalization of VSVG-KDELR with a *cis*-Golgi marker (giantin) coupled with kinetic analysis was performed. Mean with standard deviation is shown; *$p < 0.05$, paired two-tailed Student's $t$-test; $n = 3$ independent experiments. **h** HeLa cells were treated as indicated, and then colocalization of dextran with an early endosome marker (EEA1) coupled with kinetic analysis was performed. Mean with standard deviation is shown; *$p < 0.05$, paired two-tailed Student's $t$-test; $n = 3$ independent experiments

has been suggested to protect cells from osmotic stress[18]. However, mammalian cells are not typically exposed to this form of stress. Instead, because the intracellular pathways require the participation of GTPases and ATPases, which consume substantial energy, we explored whether ALDH7A1 could have a protective role during energy stress. Two major pathologic examples of energy stress are hypoxia and starvation. Initially, we found that hypoxia inhibits the identical set of transport pathways as those targeted by ALDH7A1 (Fig. 5a–h). Moreover, siRNA against *ALDH7A1* prevents these inhibitions (Fig. 5a–h). Examining starvation, we found that this condition also inhibits the identical set of pathways as those targeted by ALDH7A1, with siRNA against *ALDH7A1* similarly preventing these inhibitions (Fig. 6a–h).

We next examined whether ALDH7A1 is critical for cellular energy homeostasis during these two major states of energy deprivation. We initially found that viability is not appreciably compromised when cells are subjected to siRNA against *ALDH7A1* in the normal (non-hypoxic) condition (Fig. 7a). However, when the siRNA-treated cells are subjected to hypoxia, we observed enhanced cell death (Fig. 7a). Consistent with these findings, whereas the total ATP level is maintained when cells were treated with siRNA against *ALDH7A1* in the normal (non-hypoxic) condition, we found that ATP level drops dramatically when cells are subjected to both hypoxia and siRNA against *ALDH7A1* (Fig. 7b).

To examine starvation, we switched cells into a medium that lacks glucose and amino acids for nutrient deprivation, but

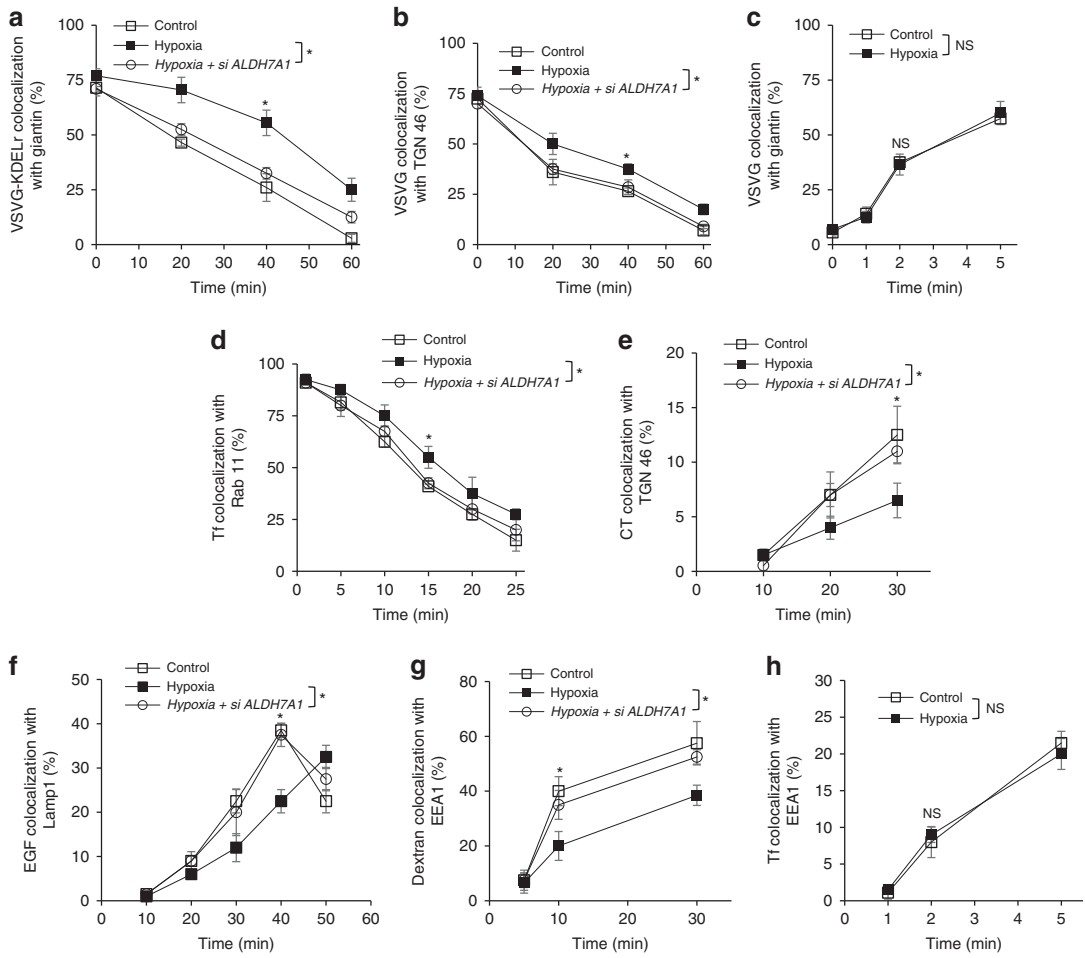

**Fig. 5** Hypoxia inhibits the identical pathways targeted by ALDH7A1. Quantitative results are shown as mean with standard deviation; *p < 0.05, NS not significant, paired two-tailed Student's t-test; n = 3 independent experiments. Source data are provided as a Source Data file. HeLa cells were assessed for the following transport pathways: **a** COPI transport from the Golgi to the ER was assessed through the quantitative colocalization of VSVG-KDELR with a cis-Golgi marker (giantin). **b** Transport of VSVG from the Golgi to the plasma membrane was assessed through the quantitative colocalization of VSVG with a TGN marker (TGN46). **c** Transport of VSVG from the ER to the Golgi was assessed through the quantitative colocalization of VSVG with a cis-Golgi marker (giantin). **d** Endocytic recycling of Tf from the recycling endosome (RE) to the plasma membrane was assessed through the quantitative colocalization of internalized Tf with a recycling endosome marker (Rab11). **e** Endocytic transport of CT to the Golgi was assessed through the quantitative colocalization of internalized CT with a TGN marker (TGN46). **f** Endocytic transport of internalized EGFR to the lysosome was assessed through the quantitative colocalization of internalized EGF with a lysosome marker (Lamp1). **g** Fluid-phase endocytosis of dextran was assessed through the quantitative colocalization of internalized dextran with an early endosome marker (EEA1). **h** Clathrin-mediated endocytosis of Tf was assessed through the quantitative colocalization of internalized transferrin (Tf) with an early endosome marker (EEA1)

contains serum for cell survival. Upon this switch, we observed the total cellular ATP level to drop immediately (Fig. 7c). Notably, this drop was magnified when the starved cells were also subjected to siRNA against *ALDH7A1* (Fig. 7c). Consistent with this result, cell death also increased when cells were subjected to both starvation and siRNA against *ALDH7A1* (Fig. 7d).

Besides HeLa cells, we also examined HEK293 cells. Upon siRNA against *ALDH7A1*, we again observed total ATP level (Supplementary Fig. 7a) and cell viability to be reduced during hypoxia (Supplementary Fig. 7b). Moreover, siRNA against *ALDH7A1* also further enhanced the drop in the total ATP level (Supplementary Fig. 7c) and cell viability (Supplementary Fig. 7d) seen during starvation.

Next, to link transport inhibition by ALDH7A1 to its ability to promote energy homeostasis, we examined the effect of expressing the mutant BARS (G172E), which prevents the BARS-dependent transport pathways from being inhibited by ALDH7A1. To express a physiologic level of the mutant BARS, we treated cells with siRNA against *BARS* followed by limited

expression of the siRNA-resistant mutant BARS (Fig. 7e). We first confirmed that the expression of the mutant BARS prevents hypoxia from exerting transport inhibition, as assessed by COPI transport (Fig. 7f) and fluid-phase endocytosis (Fig. 7g). We then found that the expression of the BARS mutant reduces total ATP level (Fig. 7h) and cell viability (Fig. 7i) during hypoxia. Similar results were also seen in starvation, as the reduced total ATP level (Fig. 7j) and cell viability (Fig. 7k) were magnified by the expression of the mutant BARS.

We next considered that AMP-activated protein kinase (AMPK) acts as a master coordinator of cellular energy homeostasis, which involves its ability to sense energy deficit and then target effectors to restore energy balance[26]. Thus, a potential role for AMPK in regulating ALDH7A1 would further support ALDH7A1 acting in energy homeostasis, as well as gaining further insights into how ALDH7A1 could be regulated in this process. Initially, we found that the inhibition of transport by hypoxia also requires AMPK, as reflected by siRNA against *AMPKα1* that prevented hypoxia from inhibiting both COPI

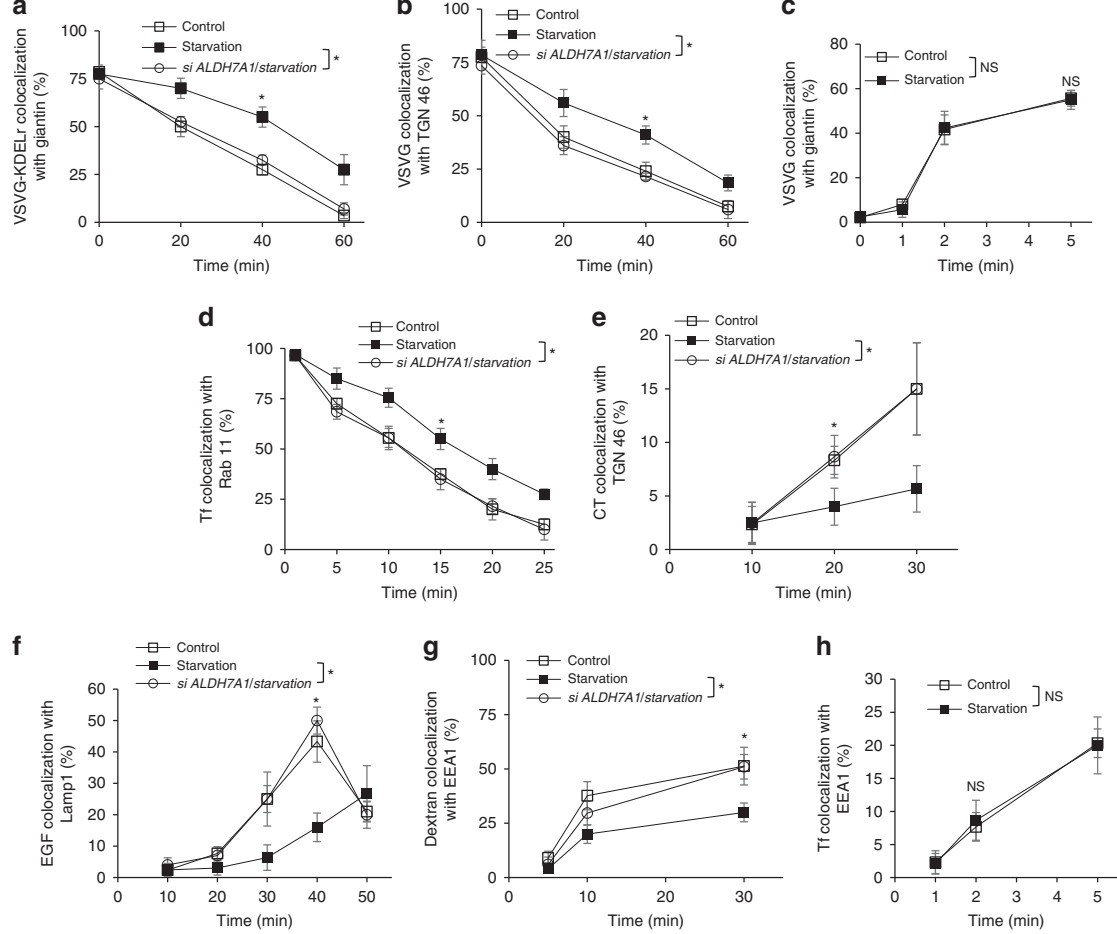

**Fig. 6** Starvation inhibits the identical pathways targeted by ALDH7A1. Quantitative results are shown as mean with standard deviation; *$p < 0.05$, NS not significant, paired two-tailed Student's $t$-test; $n = 3$ independent experiments. Source data are provided as a Source Data file. HeLa cells were assessed for the following transport pathways: **a** COPI transport from the Golgi to the ER was assessed through the quantitative colocalization of VSVG-KDELR with a cis-Golgi marker (giantin). **b** Transport of VSVG from the Golgi to the plasma membrane was assessed through the quantitative colocalization of VSVG with a TGN marker (TGN46). **c** Transport of VSVG from the ER to the Golgi was assessed through the quantitative colocalization of VSVG with a cis-Golgi marker (giantin). **d** Endocytic recycling of Tf from the recycling endosome (RE) to the plasma membrane was assessed through the quantitative colocalization of internalized Tf with a recycling endosome marker (Rab11). **e** Endocytic transport of CT to the Golgi was assessed through the quantitative colocalization of internalized CT with a TGN marker (TGN46). **f** Endocytic transport of internalized EGFR to the lysosome was assessed through the quantitative colocalization of internalized EGF with a lysosome marker (Lamp1). **g** Fluid-phase endocytosis of dextran was assessed through the quantitative colocalization of internalized dextran with an early endosome marker (EEA1). **h** Clathrin-mediated endocytosis of Tf was assessed through the quantitative colocalization of internalized transferrin (Tf) with an early endosome marker (EEA1)

transport (Fig. 8a) and fluid-phase endocytosis (Fig. 8b). Similar results were also obtained for starvation, as the inhibition of COPI transport (Fig. 8c) and fluid-phase endocytosis (Fig. 8d) by starvation also require AMPK.

We next queried the relationship between AMPK and ALDH7A1. AMPK activity can be activated in cells by pharmacologic treatment using 5-aminoimidazole-4-carboxamide-1-β-D-ribofuranoside (AICAR)[27,28]. We found that this treatment is sufficient to inhibit transport in the normal condition, as assessed by COPI transport (Fig. 8e) and fluid-phase endocytosis (Fig. 8f). We also found that transport inhibition induced by AICAR is prevented by siRNA against *ALDH7A1* (Fig. 8e, f). These effects are selective, as siRNA against *ALDH7A1* does not affect another cellular process targeted by AMPK, the activation of sirtuin 1 activity[29] (Fig. 8g). Moreover, siRNA against *AMPKα1* does not affect clathrin-mediated endocytosis (Fig. 8h), a BARS-independent pathway that is not targeted by ALDH7A1. We also found that the expression of the mutant BARS (G172E) prevents AICAR from inhibiting COPI transport (Fig. 8i) and

fluid-phase endocytosis (Fig. 8j). Thus, the collective results suggested that AMPK acts mechanistically upstream of ALDH7A1 and BARS in regulating the transport pathways.

Next, to examine how AMPK regulates ALDH7A1, we pursued an algorithm that predicts consensus sites on substrates targeted by different kinases[30]. This approach suggested two residues in ALDH7A1, serine at position 102 (S102) and serine at position 146 (S146), as potential sites of phosphorylation by AMPK. To determine whether AMPK phosphorylates ALDH7A1 at these residues, we next performed in vitro kinase assays by incubating AMPK and ALDH7A1 as purified components. To detect phosphorylation, we used an antibody that recognizes consensus sites of AMPK phosphorylation (referred hereon as the phospho-AMPK substrate antibody). We found that wild-type ALDH7A1 is phosphorylated by AMPK (Fig. 9a). Moreover, this phosphorylation is prevented when the S102 residue in ALDH7A1 is mutated to alanine (S102A), but not when the S146 residue is mutated to alanine (S146A) (Fig. 9a). Thus, these results suggested that AMPK only phosphorylates ALDH7A1 at the S102 residue.

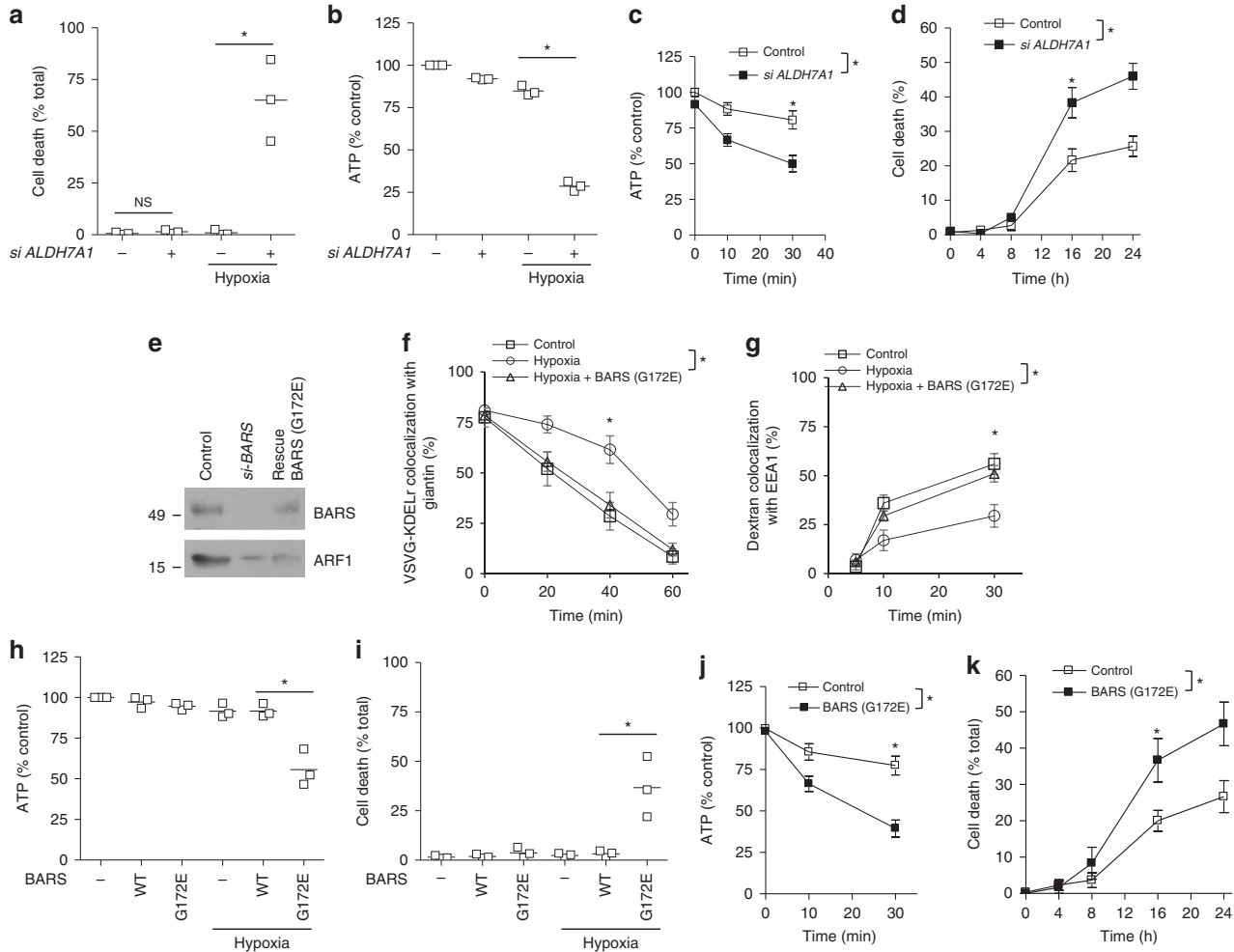

**Fig. 7** ALDH7A1 promotes energy homeostasis during hypoxia and starvation. Quantitative results are shown as mean with standard deviation; *$p < 0.05$, NS not significant, paired two-tailed Student's $t$-test. Source data are provided as a Source Data file. **a** HeLa cells were treated as indicated, and then cell death was quantified; $n = 3$ independent experiments. **b** HeLa cells were treated as indicated, and then total ATP level was quantified; $n = 3$ independent experiments. **c** Starved HeLa cells were treated as indicated, and then total ATP level was quantified; $n = 3$ independent experiments. **d** Starved HeLa cells were treated as indicated, and then cell death was quantified; $n = 3$ independent experiments. **e** Immunoblotting of HeLa cell lysates showing the efficacy of siRNA against BARS, and the level of mutant BARS expressed for rescue studies; $n = 2$ independent experiments with a representative result shown. **f** COPI transport from the Golgi to the ER in HeLa cells was assessed through the quantitative colocalization of VSVG-KDELR with a *cis*-Golgi marker (giantin); $n = 3$ independent experiments. **g** Fluid-phase endocytosis in HeLa cells was assessed through the quantitative colocalization of internalized dextran with an early endosome marker (EEA1); $n = 3$ independent experiments. **h** HeLa cells were treated as indicated, and then total ATP level was quantified; $n = 3$ independent experiments. **i** HeLa cells were treated as indicated, and then cell death was quantified; $n = 3$ independent experiments. **j** HeLa cells were treated with starvation conditions as indicated, and then total ATP level was quantified; $n = 3$ independent experiments. **k** HeLa cells were treated with starvation conditions as indicated, and then cell death was quantified; $n = 3$ independent experiments

This conclusion was supported by another approach. We found that radio-labeled phosphate is incorporated into wild-type, but not the S102A mutant, of ALDH7A1 in the kinase assay (Supplementary Fig. 8a). We also assessed the stoichiometry of this phosphorylation by tracking the rate of ADP generation during the kinase assay. When compared with a known optimal substrate of AMPK, a peptide derived from acetyl CoA carboxylase (ACC)[26], we found that AMPK phosphorylates ALDH7A1 with reasonable efficiency (Supplementary Fig. 8b and Supplementary Table 2). Thus, ALDH7A1 is likely a relevant substrate of AMPK.

We also sought to confirm this conclusion through cell-based studies. First, we found that conditions that activate AMPK (hypoxia or AICAR treatment) enhance AMPK-mediated phosphorylation of ALDH7A1 in cells (Fig. 9b). Second, this phosphorylation occurs at the S102 residue, as the enhanced

phosphorylation of ALDH7A1 induced by hypoxia is abrogated by expressing the S102A mutation, but not the S146A mutation, of ALDH7A1 (Fig. 9c).

We then examined how the phosphorylation status of ALDH7A1 regulates the transport pathways by mutating the S102 residue to alanine (S102A), which prevents its phosphorylation, or to aspartate (S102D), which mimics its constitutive phosphorylation. To avoid the overexpression of these ALDH7A1 mutants, we treated cells with siRNA against *ALDH7A1* followed by the limited expression of the transfected mutants (Fig. 9d). In cells that expressed the S102A mutant, we found that hypoxia can no longer inhibit COPI transport (Fig. 9e) or fluid-phase endocytosis (Fig. 9f). On the other hand, we found that the expression of the S102D mutant in the normal (non-energy stress) condition is sufficient to inhibit COPI transport (Fig. 9g) and fluid-phase endocytosis (Fig. 9h).

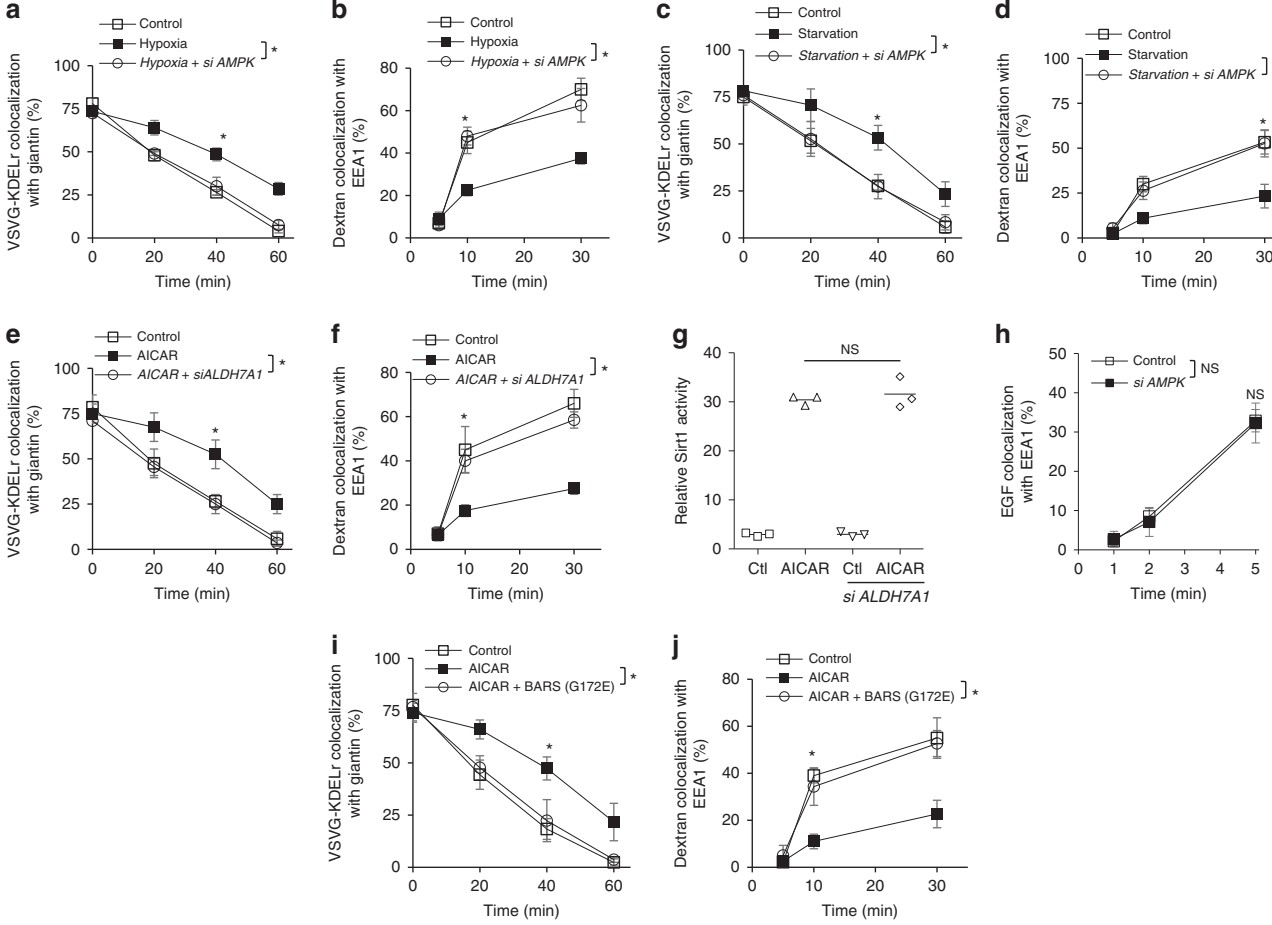

**Fig. 8** AMPK inhibits the transport pathways through ALDH7A1. Quantitative results are shown as mean with standard deviation; *$p < 0.05$, NS not significant, paired two-tailed Student's $t$-test; $n = 3$ independent experiments. Source data are provided as a Source Data file. **a** COPI transport from the Golgi to the ER in HeLa cells was assessed through the quantitative colocalization of VSVG-KDELR with a *cis*-Golgi marker (giantin). **b** Fluid-phase endocytosis of dextran in HeLa cells was assessed through the quantitative colocalization of internalized dextran with an early endosome marker (EEA1). **c** COPI transport from the Golgi to the ER in HeLa cells was assessed through the quantitative colocalization of VSVG-KDELR with a *cis*-Golgi marker (giantin). **d** Fluid-phase endocytosis of dextran in HeLa cells was assessed through the quantitative colocalization of internalized dextran with an early endosome marker (EEA1). **e** COPI transport from the Golgi to the ER in HeLa cells was assessed through the quantitative colocalization of VSVG-KDELR with a *cis*-Golgi marker (giantin). **f** Fluid-phase endocytosis of dextran in HeLa cells was assessed through the quantitative colocalization of internalized dextran with an early endosome marker (EEA1). **g** HeLa cells were treated as indicated, and then Sirt1 activity was quantified. **h** EGF endocytosis in HeLa cells was assessed through the quantitative colocalization of internalized EGF with an early endosome marker (EEA1). **i** COPI transport from the Golgi to the ER in HeLa cells was assessed through the quantitative colocalization of VSVG-KDELR with a *cis*-Golgi marker (giantin). **j** Fluid-phase endocytosis of dextran in HeLa cells was assessed through the quantitative colocalization of internalized dextran with an early endosome marker (EEA1)

The effects of the phosphorylation mutants suggested yet another way of linking transport inhibition by ALDH7A1 to its ability to promote energy homeostasis. We found that the expression of the S102A mutant results in markedly decreased cellular ATP level during hypoxia, but not in the normal condition (Fig. 9i). Cell viability is also only reduced, when the S102A mutant is expressed in conjunction with hypoxia (Fig. 9j). In contrast, the expression of the S102D mutant promotes the ability of the cell to maintain total ATP level (Fig. 9i) and viability (Fig. 9j) during hypoxia. Similar results were obtained when we examined starvation. The ability of the cell to maintain its ATP level during starvation is worsened by the expression of the S102A mutant, while the S102D mutant shows a protective effect (Fig. 9k). Moreover, expression of the S102A mutant worsens cell viability during starvation, while expression of the S102D is protective (Fig. 9l). We also expressed the G172E mutant of BARS in cells, and found that the S102D mutant of ALDH7A1 can no longer protect the total ATP level (Supplementary Fig. 8c) and cell viability (Supplementary Fig. 8d) during starvation. Thus,

when taken altogether, the effects of the ALDH7A1 phosphorylation mutants provided substantial further support that ALDH7A1 promotes in cellular energy homeostasis during situations of energy deprivation by reducing energy consumption through the inhibition of transport pathways.

**Phosphorylation by AMPK recruits ALDH7A1 to membranes**. We next examined how the phosphorylation of ALDH7A1 at its S102 residue regulates its ability to exert transport inhibition. We initially found that the catalytic activity is unaffected by mutating the S102 residue (Fig. 10a). We then considered that the recruitment from the cytosol to the membrane provides a major way of regulating transport factors. Thus, we examined whether phosphorylation of the S102 residue regulates the recruitment of cytosolic ALDH7A1 to membrane. Fractionating cells into total membrane and cytosol, we found that, compared to the wild-type form, the S102A mutant has reduced distribution on the membrane, while the S102D mutant has enhanced distribution on the

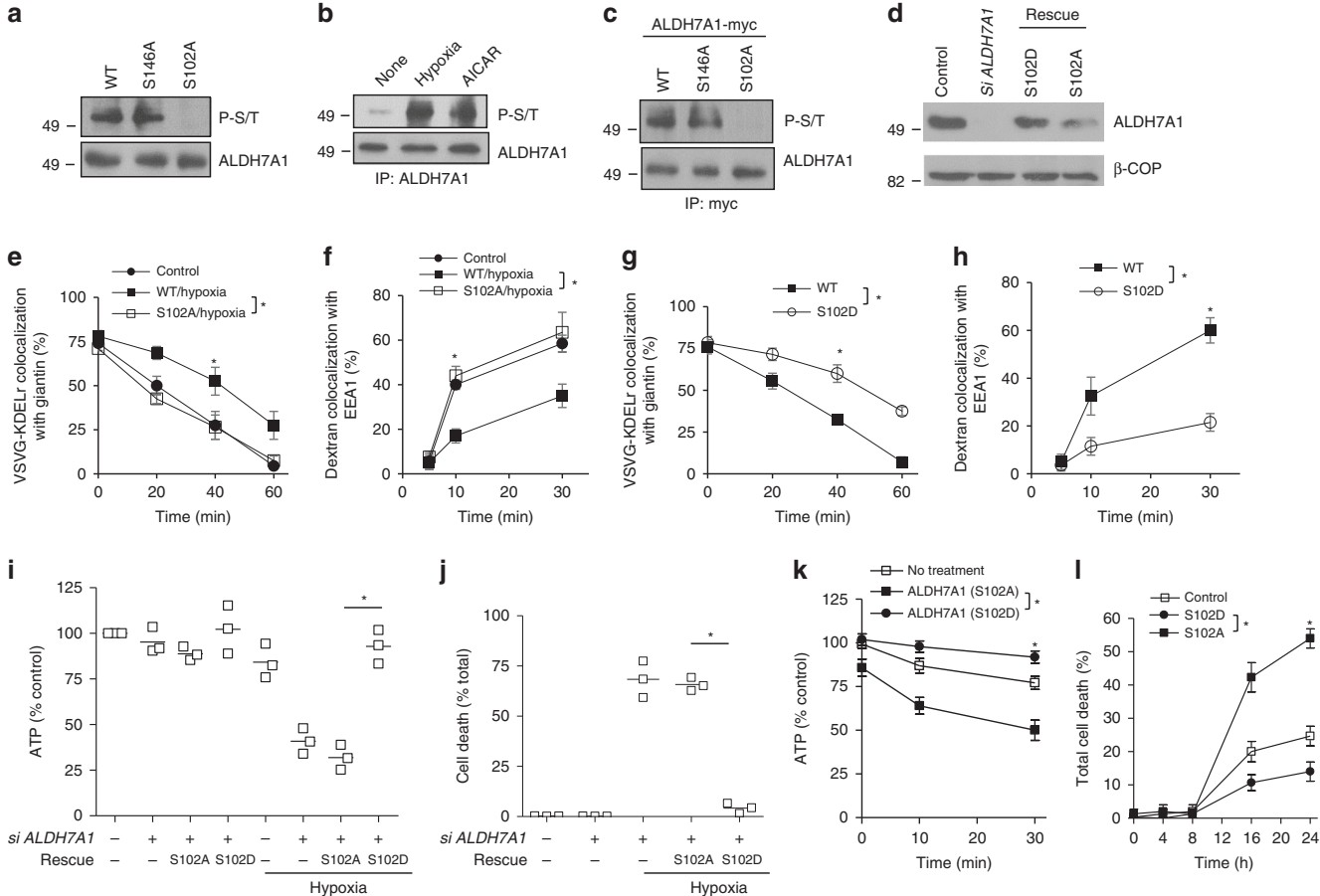

**Fig. 9** AMPK phosphorylates ALDH7A1. Quantitative results are shown as mean with standard deviation; *$p < 0.05$, NS not significant, paired two-tailed Student's $t$-test. Source data are provided as a Source Data file. **a** The in vitro kinase assay was performed, followed by immunoblotting for phospho-AMPK substrate (upper panel) or ALDH7A1 (lower panel); $n = 2$ independent experiments with a representative result shown. **b** HeLa cells were treated as indicated. ALDH7A1 was then immunoprecipitated from whole-cell lysates, followed by immunoblotting for phospho-AMPK substrate (upper panel) or ALDH7A1 (lower panel); $n = 2$ independent experiments with a representative result shown. **c** HeLa cells were transfected with constructs as indicated. Cells were than subjected to hypoxia, followed by immunoprecipitation of ALDH7A1 and then immunoblotting for ALDH7A1 or the phospho-AMPK substrate antibody; $n = 2$ independent experiments with a representative result shown. **d** Immunoblotting of whole-cell lysates showing efficacy of siRNA against *ALDH7A1*, and the level of ALDH7A1 mutants expressed for rescue studies; $n = 2$ independent experiments with a representative result shown. **e** COPI transport from the Golgi to the ER in HeLa cells was assessed through the quantitative colocalization of VSVG-KDELR with a *cis*-Golgi marker (giantin); $n = 3$ independent experiments. **f** Fluid-phase endocytosis of dextran in HeLa cells was assessed through the quantitative colocalization of internalized dextran with an early endosome marker (EEA1); $n = 3$ independent experiments. **g** COPI transport from the Golgi to the ER in HeLa cells was assessed through the quantitative colocalization of VSVG-KDELR with a *cis*-Golgi marker (giantin); $n = 3$ independent experiments. **h** Fluid-phase endocytosis of dextran in HeLa cells was assessed through the quantitative colocalization of internalized dextran with an early endosome marker (EEA1); $n$ q= 3 independent experiments. **i** HeLa cells were treated as indicated, and then total ATP level was quantified; $n = 3$ independent experiments. **j** HeLa cells were treated as indicated, and then cell death was quantified; $n = 3$ independent experiments. **k** HeLa cells were treated with starvation conditions as indicated, and then total ATP level was quantified; $n = 3$ independent experiments. **l** HeLa cells were treated with starvation conditions as indicated, and then cell death was quantified; $n = 3$ independent experiments

membrane (Fig. 10b). To ascertain that the phosphorylation of the S102 residue promotes membrane recruitment, rather than inhibiting membrane release, we incubated the phosphorylation mutants as recombinant proteins with Golgi membrane, and found that the S102D mutant becomes membrane-bound after the incubation, while the S102A mutant remains soluble (Fig. 10c).

Complementing the above results, we found by subcellular fractionation that hypoxia and starvation, conditions that activate AMPK phosphorylation of ALDH7A1, enhance the membrane distribution of ALDH7A1 (Supplementary Fig. 9a). Moreover, confocal microscopy revealed that the localization of ALDH7A1 to multiple intracellular compartments is enhanced in these conditions (Supplementary Fig. 9b). Notably, no enhancement is

observed for the ER, which is consistent with our findings above that starvation and hypoxia do not inhibit transport from the ER.

To gain further insight into how membrane recruitment promotes transport inhibition by ALDH7A1, we next considered that an isoform of nicotinamide mononucleotide adenylyltransferase (NMNAT), which catalyzes the synthesis of NAD from NMN (nicotinamide mononucleotide), has been found to exist at the Golgi[31]. We first confirmed that Golgi membrane contains an appreciable level of NAD (Fig. 10d). We then found that simply incubating recombinant ALDH7A1 with Golgi membrane results in this NAD being converted to NADH (Fig. 10d). Notably, this result suggested that Golgi membrane contains not only NAD, but also substrates of ALDH7A1, as its catalytic activity requires both substrate and cofactor to drive the reaction.

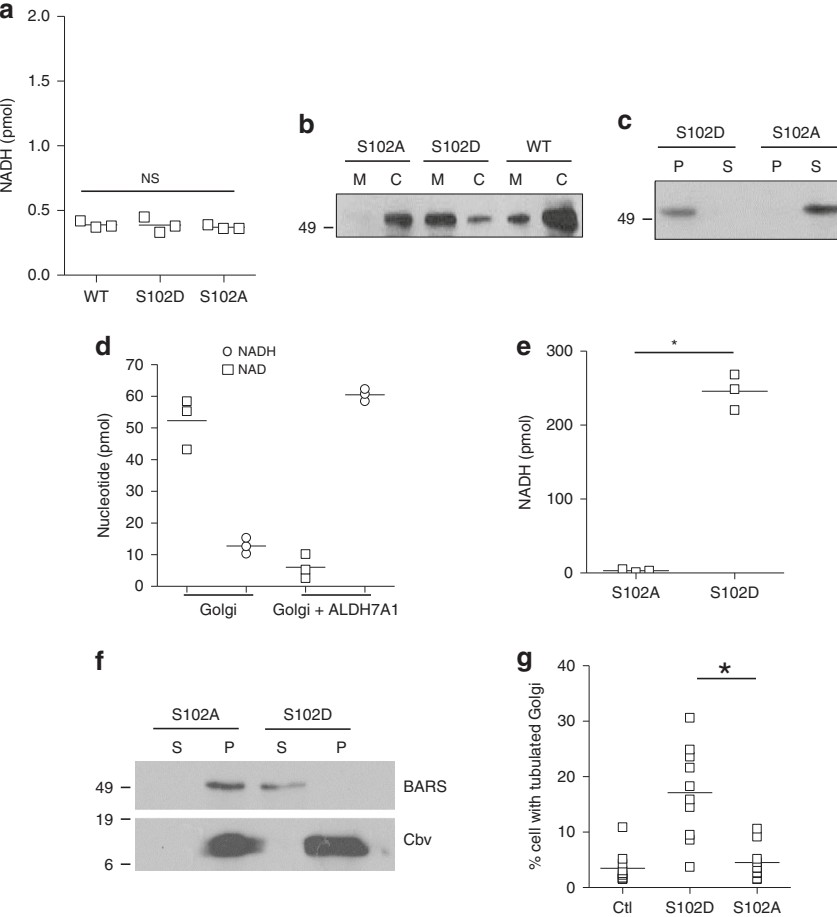

**Fig. 10** Phosphorylation of ALDH7A1 regulates its membrane recruitment. Source data are provided as a Source Data file. **a** Recombinant ALDH7A1 was incubated with a known substrate (octanal) and NAD, and the catalytic activity was tracked through the generation of NADH. Mean with standard deviation is shown; NS not significant, ANOVA; $n = 3$ independent experiments. **b** HeLa cells were transfected with different myc-tagged ALDH7A1 constructs as indicated, followed by subcellular fractionation to obtain total cytoplasmic membrane (M) and cytosol (C), and then immunoblotting using anti-myc antibody; $n = 2$ independent experiments with a representative result shown. **c** Recombinant forms of ALDH7A1 mutants were incubated with Golgi membrane, followed by centrifugation to obtain membrane (P) vs. soluble (S) fractions, and then immunoblotting for ALDH7A1; $n = 2$ independent experiments with a representative result shown. **d** Golgi membrane were treated as indicated, and then the levels of NAD and NADH on the membrane were quantified; $n = 3$ independent experiments with a representative result shown. **e** Liposomes were generated using purified lipids to mimic the composition of Golgi membrane. A substrate of ALDH7A1 (octanal) and NAD were then added, followed by incubation with different forms of recombinant ALDH7A1 as indicated. The level of NADH was then quantified. Mean with standard deviation is shown; *$p < 0.05$, paired two-tailed Student's $t$-test; $n = 3$ independent experiments. **f** Golgi membrane was incubated with mutant forms of ALDH7A1 as indicated, followed by centrifugation to re-isolate Golgi membrane (P, pellet) vs. soluble (S) fraction. Both fractions were then immunoblotted for endogenous BARS; $n = 2$ independent experiments with a representative result shown. **g** HeLa cells that expressed the different mutant forms of ALDH7A1 as indicated, achieved by siRNA against endogenous ALDH7A1 followed by transfection of mutant forms, were examined by immunofluorescence microscopy that tracked VSVG leaving the TGN. Cells that showed VSVG-positive Golgi tubules were then quantified; $n = 3$ independent experiments

We next considered that native membrane is complex, containing a myriad of membrane-bound proteins that could potentially mediate an indirect way by which ALDH7A1 converts NAD on Golgi membrane to NADH. Thus, to confirm that ALDH7A1 on the membrane can directly convert NAD to NADH, we generated liposomes with a lipid composition that mimics the lipid profile of the Golgi membrane. To this mix, we added known aldehyde substrates of ALDH7A1 along with NAD. Upon the incubation of recombinant ALDH7A1 with these liposomes, we observed the generation of NADH, and notably, this generation exhibits physiologic specificity, as only the S102D mutant induces NADH generation, but not the S102A mutant (Fig. 10e). We also confirmed that the reconstituted ALDH7A1 activity requires the presence of both NAD and substrate (Supplementary Fig. 10a). We then performed enzyme kinetics, which revealed that the presence of liposomes enhances ALDH7A1 activity by nearly 100-fold (Supplementary Table 3).

We also further scrutinized how NADH generated by ALDH7A1 inhibits BARS. When the Golgi membrane was incubated with the S102A mutant of ALDH7A1, which has reduced capacity to generate NADH (see Fig. 10e), we observed BARS to remain on the membrane (Fig. 10f). In contrast, when the Golgi membrane was incubated with the S102D mutant, which generates elevated level of NADH (see Fig. 10e), we observed BARS to become released from the membrane (Fig. 10f). Gel filtration revealed that the release BARS exists as a dimer (Supplementary Fig. 10b). We also considered that BARS has been shown to induce the fission of tubular carriers that emanate from the trans-Golgi network (TGN) for transport to the PM[23]. We found that the expression of the S102D mutant of ALDH7A1 prevents this fission role of BARS, while the S102A mutant had no appreciable effect (Fig. 10g and

Supplementary Fig. 10c). Thus, the collective results further supported that phosphorylation of the S102 residue in ALDH7A1 promotes its role in transport inhibition by enhancing its recruitment from the cytosol to the membranes.

## Discussion

ALDH7A1 activity is currently known to act in lysine metabolism[32] and osmotic stress[18]. In this study, we find that situations of energy stress induce AMPK to phosphorylate ALDH7A1, which results in its redistribution from the cytosol to intracellular membrane compartments to exert a broad inhibition of the intracellular transport pathways. Studying COPI transport, we have further elucidated that rather than the metabolic conversion of aldehydes to carboxylates, it is the reductive consequence of this reaction, which generates NADH from NAD, that underlies how ALDH7A1 inhibits COPI transport, as NADH generated by ALDH7A1 targets BARS to inhibit COPI vesicle fission.

Many metabolic enzymes generate NADH from NAD through their dehydrogenase activity. However, whereas the NADH generated by these enzymes is typically cytosolic, we find that ALDH7A1 also generates membrane-bound NADH. Thus, because BARS only interacts with ALDH7A1 on membranes, these observations suggest how ALDH7A1 targets BARS with efficiency and specificity. We further note that, whereas NADH is well known to act in cellular energetics through its participation in glycolysis and mitochondrial oxidative phosphorylation, we have uncovered a fundamentally different way that NADH generated by ALDH7A1 on membranes promotes energy homeostasis, by reducing energy consumption through transport inhibition.

Our findings also advance a mechanistic understanding of COPI vesicle formation. Early studies identified a major mechanism of regulation, which involves the small GTPase ARF1 regulating the recruitment of coatomer, the core components of the COPI complex, from the cytosol to membranes[9]. In the current study, we find that ALDH7A1 regulates the fission stage of COPI vesicle formation. We further note that we have recently identified another metabolic enzyme, GAPDH, also to inhibit COPI vesicle fission[19]. Whereas ALDH7A1 targets BARS, GAPDH targets ARFGAP1. Thus, the collective findings from these two studies suggest that the fission stage represents another major node by which COPI transport can be targeted for regulation.

It is further notable that the intracellular transport pathways have not been known to be a major target by which significant reduction in cellular energy consumption can be achieved. An explanation is suggested by the consideration that studies on vesicular transport have typically focused on single pathways. In contrast, we have found that, by coordinating inhibition across multiple pathways, a meaningful reduction in energy consumption can be achieved to promote energy homeostasis during situations of energy stress. This explanation is further supported by our recent elucidation that GAPDH also relieves energy stress by exerting a broad inhibition of the transport pathways[19].

Besides having roles in lysine metabolism[32] and osmotic stress[18], ALDH7A1 activity has also been suggested to protect the cell against lipid peroxidation[22,33], which is a form of oxidative stress that occurs in multiple pathologic conditions[34]. As we have elucidated that transport inhibition by ALDH7A1 requires its recruitment to membranes, and lipid peroxidation also occurs on membranes, an intriguing possibility is that ALDH7A1 activity on membranes coordinates two general processes to protect against two major types of cellular stress: (i) inducing transport inhibition to reduce energy consumption in protecting against energy stress, which is mediated through the generation of NADH, and (ii) reducing lipid peroxidation to protect against oxidative stress, which is mediated through the metabolic conversion of aldehydes to carboxylates. This possibility predicts that ALDH7A1 activity may be particularly important for cell survival in pathologic conditions when both types of stresses are present.

We further note that hereditary deficiency in ALDH7A1 activity has been identified[32]. As we have found that ALDH7A1 is not critical under normal (non-stressed) conditions, this finding explains how such humans can exist. However, there are situations when organs of the human body can encounter substantial hypoxia and starvation. A major pathologic example is cardiac ischemia. As such, our finding that ALDH7A1 protects the cell against situations of energy deprivation not only predicts that cardiac ischemia should be more devastating for patients with deficient ALDH7A1 activity, but also suggests a therapeutic target for the treatment of this major cause of human morbidity and mortality.

## Methods

**Chemicals and proteins**. NAD, NADH, AICAR, hexanoic acid, octanoic acid, octanal, betaine aldehyde, betaine, and 2-aminoadipic acid were obtained (Sigma). Cholesterol, sphingomyelin, DOPC, DOPE, DOPS, and phosphatidic acid (PA) were obtained (Avanti). Alexa 546-conjugated transferrin (Tf), Alexa 555-conjugated forms of EGF, dextran, and CT were also obtained (Invitrogen). Fugene 6 Transfection Reagent was from Promega. RNAiMax was from Invitrogen. Preparations of coatomer, ARF1, ARFGAP1, BARS, Golgi membrane, and cytosol have been described[14,16]. To generate recombinant ALDH7A1, the human cDNA was expressed in bacteria (BL21) using the pET-15b vector (Novagen), followed by isolation of the 6×-his tagged form of ALDH7A1 using a nickel column. AMPK was purified from cells by transfection of a flag-tagged construct into HeLa cells, followed by immunoprecipitation of the tagged construct from cell lysates using an anti-flag antibody (M2).

**Cells**. HeLa and HEK293 cells were obtained from American Type Culture Collection (ATCC), which were documented to be free of mycoplasma contamination. Cells were cultured in Dulbecco's Modified Eagle Medium (DMEM) with 10% fetal bovine serum and supplemented with glutamine. Starvation medium consisted of phosphate-buffered saline supplemented with 10% dialyzed bovine serum. Hypoxia studies were performed by incubating cells in a hypoxia chamber (filled with premixed air that contains 1% oxygen) at 37 °C for 16 h.

**Plasmids**. Mammalian expression vectors, which contain temperature-sensitive mutations (ts-045) of VSVG and VSVG-KDELR, have been described[24]. The BARS mutant (G172E) has been described previously[16], and was subcloned into pcDNA3.1 for transfection studies. Flag-AMPK in pcDNA3 was a gift from Dr. Bing Zheng (Columbia University, NY). ALDH7A1 was subcloned into mammalian expression vectors, pcDNA3.1 and pEGFP-N1. Mutant ALDH7A1 (E268Q) was generated using QuikChange Site-Directed-Mutagenesis (Stratagene) with paired oligonucleotides: 5′-gggagaagtctgttgcaacttggaggaaacaatgcc-3′ and 5′-ggcattgtttcctccaagttgcaacagacttctccc-3′. Mutant ALDH7A1 (S102A) was generated with paired oligonucleotides: 5′-aagatccaagtactaggagccttggtgtctttggagatgggg-3′ and 5′-ccccatctccaaagacaccaaggctcctagtacttggatctt-3′. Mutant ALDH7A1 (S102D) was generated with paired oligonucleotides: 5′-aagatccaagtactaggagacttggtgtctttgga-gatgggg-3′ and 5′-ccccatctccaaagacaccaagtctcctagtacttggatctt-3′. Mutant ALDH7A1 (S146A) was generated with paired oligonucleotides: 5′-atcttgccttctgaaa-gatctggccatgcactgattgag-3′ and 5′-ctcaatcagtgcatggccagatctttcagaaggcaagat-3′.

**Antibodies**. Rabbit antibody against human ALDH7A1 (WB: 1:1000, IF: 1:500) was generated by injecting the recombinant protein into rabbits using a commercial service (Proteintech). The following antibodies have been described in our previous studies[14,16,24,35]: ARF1 (WB 1:1000), BARS (WB 1:1000), coatomer (IF 1:10), β-COP (WB 1:10), calnexin (IF 1:200), cellubrevin (WB 1:1000), giantin (IF 1:2000), GM130 (IF 1:500), Lamp1 (IF 1:200), Sec61p (IF 1:500), TGN46 (IF 1:200), VSVG (IF 1:10), HA epitope tag (WB 1:200), and Myc epitope tag (WB 1:200). The following antibodies were obtained from commercial sources: actin (WB 1:1000, Ambion, AM4302), EEA1 (1:100 IF, BD Transduction Laboratories, 610456), GFP (WB 1:1000, Thermo Scientific, MA5-15256), GST (WB 1:1000, Santa Cruz Biotechnology, sc-138), and Rab11 (IF 1:200, BD Biosciences, 610656). Other primary antibodies were obtained from Cell Signaling: ALDH1A1 (12035S, WB 1:1000), AMPKα1 (2603S, WB 1:500), phospho-substrate-AMPK antibody (5759S, WB 1:500), Lamin B (12586S, WB 1:1000), and Sirt1 (2493S, WB 1:500). Conjugated secondary antibodies were obtained from Jackson ImmunoResearch: horseradish peroxidase-conjugated donkey antibodies against mouse IgG (715-035-150, WB 1:10,000) and against rabbit IgG (711-035-152, WB 1:10,000), Cy2 donkey antibodies against mouse IgG (715-225-151, IF 1:200) and against rabbit IgG (711-225-152, IF 1:200), Cy3 goat antibody against mouse IgG (115-165-062, IF 1:200),

Cy3 donkey antibodies against rabbit IgG (711-165-152, IF 1:200) and against sheep IgG (713-165-147, IF 1:200).

**Mass spectrometry.** Cytosol was purified from rat livers by homogenization in lysis buffer (1 mM Tris pH 7.4, 800 mM sucrose, 5 mM EDTA, and protease inhibitors) at 4 °C, followed by centrifugation at 80,000×$g$ for 90 min. The resulting supernatant was dialyzed against dialysis buffer (25 mM Tris at pH 8.0, 50 mM KCl, and 1 mM DTT), followed by centrifugation at 150,000×$g$ for 90 min at 4 °C. Aliquots were stored at −80 °C. Pulldown experiment was performed by incubating GST or GST-BARS on beads (5 µg) with 50 µg rat liver cytosol in traffic buffer (25 mM HEPES pH 7.2, 50 mM KCl, and 2.5 mM Mg(OAc)₂) for 1 h followed by extensive washing. Samples were separated by SDS-PAGE and then stained with Coomassie blue. Specific bands were excised for protein identification by mass spectrometry (Taplin Mass Spectrometry Facility, Harvard Medical School, USA). Interacting proteins were identified for both GST and GST-BARS. Proteins that only interacted with GST-BARS were considered specific.

**Transfections and siRNA.** Transfection of DNA plasmids was performed using FuGene6 (Roche). Transfection of siRNA was performed using Lipofectamine RNAiMAX (Invitrogen). Plasmid transfections occurred over 48 h and siRNA transfections occurred over 72 h. The siRNA sequences used to target human *ALDH7A1*, 5′-gcagugagcauguuucuugtt-3′ (sense strand) and 5′-caagaaacaugcuca cugctt-3′ (complement strand), were obtained (Dharmacon). Rescue plasmids for wild-type and catalytic dead mutant of human ALDH7A1 were generated by targeting this siRNA sequence using QuikChange Site-Directed-Mutagenesis (Stratagene) with paired oligonucleotides: 5′-cacaccaagcaggcagtgtcgatgtttcttggag cagtg-3′ and 5′-cactgctccaagaaacatcgacactgcctgcttggtgtg-3′. Oligonucleotides that target human *AMPKα1*, 5′-agugaagguuggcaaacautt-3′ (sense strand) and 5′-auguuugccaaccuucacutt-3′ (complement strand), human *ALDH1A1*, 5′-guagccuu cacaggaucaauu-3′ (sense strand) and 5′-uugauccugugaaggcuacuu-3′ (complement strand), human *catalase*, 5′-uggauauggaucacauacu-3′ (sense strand) and 5′-aguaugugauccauaucca-3′ (complement strand), human *Ftcd*, 5′-ccaaucuucugga cuuuga-3′ (sense strand) and 5′-ucaaaguccagaagauugg-3′ (complement strand), human *malate dehydrogenase 2*, 5′-uggugagccgccugacccu-3′ (sense strand) and 5′-agggucaggcggcucacca-3′ (complement strand), and human *aldolase*, 5′-ccgagaa caccgaggagaa-3′ (sense strand) and 5′-uucuccucggguuucucgg-3′ (complement strand) were also obtained (Dharmacon). Targeting specificities of these siRNAs has been documented previously[36,37].

**In vivo transport assays.** Quantitative microscopy-based transport assays of the different intracellular pathways has been described previously[24]. In brief, they are done as follows.

For anterograde transport from ER to Golgi, cells were transfected with pROSE-VSVG-ts045-Myc for 1 day, and then incubate at 39 °C for 4 h to accumulate VSVG in the ER. Cells were then shifted to 32 °C for different times as indicated in figures. Cells were then stained for giantin, followed by confocal microscopy to assess the arrival of VSVG to the Golgi.

For retrograde transport from the Golgi to the ER, cells were transfected with pROSE-VSVG-ts045-KDELR-Myc for 1 day, and then incubated at 32 °C for 8 h to achieve steady-state distribution at the Golgi. Cells were then shifted to 39 °C for different times as indicated in the figures. Cells were then stained for giantin, followed by confocal microscopy to assess the exit of VSVG-KDELR from the Golgi.

For anterograde transport from the Golgi to the PM, cells were transfected with pROSE-VSVG-ts045-Myc for 1 day, and then incubated at 20 °C for 2 h to accumulate VSVG at the TGN. Cells were then shifted to 32 °C for different times as indicated in the figures. Cells were then stained for TGN46, followed by confocal microscopy to assess the exit of VSVG from the Golgi.

For the recycling of Tf from the early recycling endosome to the PM, cells were incubated with Alexa 546-conjugated Tf (5 µg/ml in DMEM) at 37 °C for 2 h to allow the steady-state accumulation of Tf in endosomes. Subsequently, cells were incubated in a medium without Tf for different times as indicated in the figures. Cells were then stained for Rab11, followed by confocal microscopy to assess the exit of Tf from the early recycling endosome.

For the retrograde transport of CT from the PM to the Golgi, cells were incubated with Alexa 555-conjugated CT for 30 min at 4 °C (0.5 µg/ml in DMEM). After washing to clear unbound CT, cells were shifted to 37 °C for different times as indicated in the figures. Cells were then stained for TGN46, followed by confocal microscopy to assess the arrival of CT to the Golgi.

For endocytic transport of EGF to the lysosome, cells were incubated with Alexa 555-conjugated EGF (1 µg/ml in DMEM) for 1 hour at 4 °C. Cells were then washed to clear unbound EGF, followed by shifting to 37 °C for times indicated in the figures. Cells were stained for Lamp1, followed by confocal microscopy to assess the arrival of EGF to the lysosome.

For the fluid-phase uptake of dextran, cells were incubated with Alexa 555-conjugated dextran (0.2 mg/ml) at 37 °C for different times as indicated in the figures. Cells were then stained for EEA1, followed by confocal microscopy to assess the arrival of dextran to the early endosome.

For the endocytosis of EGF, cells were incubated with Alexa 555-conjugated EGF (1 µg/ml in DMEM) for 1 h at 4 °C. Cells were then washed to clear unbound EGF, followed by shifting to 37 °C for times indicated in the figures. Cells were stained for EEA1, followed by confocal microscopy to assess the arrival of EGF to the early endosome.

**Confocal microscopy.** Colocalization studies were performed using two confocal systems. The Nikon system is equipped with the Nikon Eclipse TE2000U Inverted Microscope having a Plan Apo 60×/1.40 oil objective, Nikon D-Eclipse C1 confocal package with a 488 Laser (having 515/30 emission filter) and a 543 Laser (having 590/50 emission filter), and Nikon EZ-C1 version 3.90 acquisition software. The Zeiss system is equipped with the Zeiss Axio Observer Z1 Inverted Microscope having a Plan-Apochromat 63× objective, the Zeiss LSM 800 with Airyscan confocal package with Zeiss URGB (488 and 561 nm) laser lines, and Zen 2.3 blue edition confocal acquisition software.

For quantitation of colocalization, ten fields of cells were examined, with each field typically containing five cells. Images were imported into the NIH Image J version 1.50e software, and then analyzed through a plugin software (https://imagej.net/Coloc_2). Under the "Image" tab, the "Split Channels" option was selected. Under the "Plugins" tab, "Colocalization Analysis" option was selected, and within this option, the "Colocalization Threshold" option was selected. Colocalization values were then calculated by the software, and expressed as the fraction of protein of interest (cargo or ALDH7A1) colocalized with an organelle marker.

**In vitro reconstitution of COPI vesicle formation.** The reconstitution system was performed essentially as previously described[14,16]. Briefly, Golgi membrane (0.2 mg/ml) was washed with 3 M KCl, and then incubated with ARF1 (6 µg/ml) and coatomer (6 µg/ml) for the first-stage incubation that reproduces the ARF-dependent recruitment of coatomer onto the Golgi membrane. The Golgi membrane was re-isolated and then incubated with ARFGAP1 (2 µg/ml) and BARS (2 µg/ml) for the second stage that results in vesicle formation. ALDH7A1 (1 µg/ml) was added at the second stage. Products of ALDH7A1 activity were also added at this stage. These include: hexanoic acid (20 µM), octanoic acid (20 µM), 2-aminoadipic acid (20 µM), or NADH (20 µM).

EM examination of Golgi membrane using the whole-mount technique has been described previously[14,16]. In brief, membrane samples from the COPI reconstitution system were spotted onto EM grids and then fixed with 2% PFA/PBS for 10 min. Grids were rinsed three times with water, followed by 1% uranyl acetate staining, and then examined using JEOL 1200EX Transmission electron microscope.

**Subcellular fractionation.** Cells were washed with PBS, resuspended in homogenization buffer (0.25 M sucrose, 1 mM EDTA, and 20 mM HEPES-KOH, pH 7.4 and protease inhibitor cocktail) and then disrupted by passing through 28-gauge needles. After low-speed centrifugation (800×$g$ for 6 min) to spin out nuclei and unbroken cells, the resulting post-nuclear supernatant was centrifuged at 100,000×$g$ for 1 h to obtain cytosol and total membrane fractions.

To obtain nuclear vs. cytoplasmic fractions, cells were washed with PBS and resuspended into hypotonic buffer containing 10 mM HEPES, pH 8.0, 10 mM KCl, 3 mM MgCl₂, 0.5 mM DTT, and protease inhibitors. The suspension was then incubated on ice for 10 min and Triton X-100 was added to a final concentration of 0.3%. After low-speed centrifugation (800×$g$ for 5 min), the supernatant was collected as the cytoplasmic fraction. The pellet was washed twice with hypotonic buffer again and resuspended in the lysis buffer containing 100 mM Tris-HCl, pH 8.0, 1% Triton X-100, 100 mM NaCl, 0.5 mM EDTA, and protease inhibitors. After centrifugation at 15,000×$g$ for 10 min, the supernatant was collected as the nuclear fraction.

**In vitro studies using liposomes.** Liposomes were generated from pure defined lipids to mimic the composition of native membrane (mol%): dioleoyl-phosphatidylcholine (DOPC) 50%, dioleoyl-phosphatidylethanolamine (DOPE) 16%, dioleoyl-phosphatidylserine (DOPS) 5%, PA 5%, cholesterol 17%, and sphingomyelin 7%. These pure lipids were mixed in a glass tube, dried under nitrogen, and then resuspended in buffer (25 mM Hepes–KOH, pH 7.2, 50 mM KCl, 2.5 mM Mg(OAc)₂, and 0.2 M sucrose), which involved ten times of rapid freeze and thaw with occasional vortexing. The generated giant liposomes were then passed through a mini-extruder (400-nm filter) 21 times.

To reconstitute ALDH7A1 activity in the context of liposomes, liposomes (20 µg) were mixed with NAD (10 µM) and substrate (octanal (10 µM) or betaine aldehyde (10 µM)) in 200 µl of PBS, and then recombinant ALDH7A1 (1 µg) was added. Incubation was performed at 37 °C for 1 h. Subsequently, NADH measurement was performed using the enzyme cycling assay. Enzyme kinetics of ALDH7A1 were determined as previously described[22]. To calculate Km, incubation was performed by varying the level of substrate in the presence of excess NAD.

**In vitro kinase assay.** The kinase assay was performed essentially as previously described[19]. For phosphorylation detection using the AMPK phospho-substrate antibody, myc-tagged AMPK1 was isolated from transfected cells that had been

starved and treated with AICAR. For phosphorylation detection using $\gamma P^{32}$-ATP or using the ADP-Glo Kinase assay (Promega), recombinant active AMPK was obtained (Sinobiological). In all cases, AMPK was incubated with recombinant ALDH7A1 in the presence of ATP and phosphatase inhibitors. To assess the stoichiometry of phosphorylation, the SAMS peptide, a sequence from an optimal AMPK substrate, ACC, was used for comparison.

**Other assays**. Total cellular ATP level was detected using ATPlite Luminescence Detection Assay System (Perkin Elmer), and was performed according to manufacturer's instructions, with final values normalized to number of viable cells. Cell viability was assessed by incubating cells with propidium iodine (1 ng/µl) and then quantified by flow cytometry. NAD/NADH measurements were performed using the standard enzyme cycling assay according to manufacturer's instruction (BioVision).

**mRNA quantification**. Total RNA was isolated using the PureLink kit (Ambion). Total mRNA was then isolated using the MicroPolyAPurist kit (Ambion). Both were done according to manufacturer's instructions. Quantitation was performed by measuring optical density at 260 nm and normalizing for cell number. Quantitation of RNA transcript using real-time PCR (qPCR) has been described[38]. The cycling conditions were: denaturation at 95 °C for 30 s, annealing at 56 °C for 60 s, and extension at 72 °C for 30 s. Primer sequences used to detect the mRNA for E-cadherin, 5′-tcgacacccgattcaaagtgg-3′ (forward) and 5′-gtgggttatgaaaccgtagagg-3′ (reverse), for *ALDH7A1*, 5′-tttccctgtggcagtgtatg-3′ (forward) and 5′-cctcca-gaaccttggctattatc-3′ (reverse), for *Ftcd*, 5′-aacctgctcggcacaaa-3′ (forward) and 5′-ctgagccaggttcttctcatc-3′ (reverse), for *Catalase*, 5′-ctccactgttgctggagaat-3′ (forward) and 5′-cgagatcccagttaccatcttc-3′ (reverse), for *ALDO*, 5′-cctcttccatgagacactctac-3′ (forward) and 5′-cgcccttgtctaccttgat-3′ (reverse), and for *MDH2*, 5′-agccacttt cacttcctcctg-3′ (forward) and 5′-tttggtctcgatgtggctc-3′ (reverse), were purchased from Integrated DNA Technologies, Inc. (IDT).

**Gel filtration**. Analysis was performed using the AKTA UPC-900 FPLC system (GE Healthcare) with a Superdex 200 10/200GL column, with the following parameters: sample volume 500 µl, eluent buffer 20 mM Tris-HCl, 100 mM NaCl, pH 7.2, flow rate 0.5 ml/min.

**Statistical analysis**. Sample size is noted in the figure legends. Sample size used was based on our previous familiarity with the assays. For comparison between two conditions, significance was tested by the paired two-tailed Student's *t*-test. For comparison among multiple conditions, significance was tested by the analysis of variance (ANOVA). These tests were performed using Excel or Prism software. No inclusion/exclusion criteria were pre-established. The experiments were not randomized. The investigators were not blinded to the group allocation during experiments and in outcome assessment.

**Reporting Summary**. Further information on research design is available in the Nature Research Reporting Summary linked to this article.

## Data availability

The data that support the findings of this study are available from the corresponding author upon reasonable request. A reporting summary for this Article is available as a Supplementary Information file. The source data for Figs. 1a–j, 2a–g, 2i, j, 3a–i, 4a, 4c, 4e–h, 5a–h, 6a–h, 7a–k, 8a–j, 9a–l, 10a–g, Supplementary Figs. 1a–d, 2d, 3g–i, 4a–i, 5a–i, 6a–i, 7a–d, 8a–d, 9a, b, 10a, b, and Supplementary Tables 2 and 3 are provided as a Source Data file. Uncropped scans of immunoblots are also provided in this Source Data file. The mass spectrometry proteomics data have been deposited to the ProteomeXchange Consortium via the PRIDE[39] partner repository with the data set identifier PXD014645.

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

## Acknowledgements

We thank Joao Paulo and Steve Gygi for advice and discussion. This work is supported by grants from the National Institutes of Health to V.W.H. (GM058615 and GM115683) and to I.C.H. (AR070171), from the Brigham and Women's Hospital to J.S.Y. (the Evergreen Innovation Fund), and from the Department of Defense to I.C.H. (W81XWH-11-1-0492).

## Author contributions

J.S.Y., J.W.H., S.Y.P., S.Y.L., J.L., M.B., C.A., W.T., X.M., and I.C.H. performed the experiments. All authors participated in analyzing the data. V.W.H. supervised the work and wrote the manuscript with help from J.S.Y.

## Additional information

**Competing interests:** The authors declare no competing interests.

