## [Transparent Peer Review File · Nature Communications]

Reviewers' comments:

Reviewer #1 (Remarks to the Author):

This manuscript describes a possible involvement of ALDH7A1 (authors use ATQ, which is not an official and HGNC approved gene name) in a broad inhibition of the intracellular transport pathways. This is an interesting study and a number of experiments support in general their conclusion. However, all data are in one transformed cell line and there is a lack of a proposed mechanism regarding the substrates that are metabolized by ALDH7A1 and how they can contribute to the observed phenotype.

There are several issues with this manuscript.

1. The approved name of the gene/protein (HGNC) is ALDH7A1 and this should be used throughout the manuscript. The use of the alias ATQ creates confusion. The same thing for ALDH1. Is this ALDH1A1 as it is noted in one for the figures? ALDH1 subfamily consists of many genes including ALDH1A1, 1A2, 1A3, 1B1, 1L1 and 1L2. To avoid confusion in the literature and to be consistent with the journals policy, HGNC approved symbols should be used.

2. The authors claim that the NADH produced by ALDH7A1 is the major player for the function of the this gene product. However there is not any mentioning about the substrates metabolized by this protein, as well as their importance.

3. It is a little bit confusing when it comes to the cellular localization of the effects. Are they membrane specific? do they occur in the cytosol? Is this phenotype related to the nucleus? Brocker et al, have shown that this ALDH7A1 gene is expressed in mitochondria, cytosol and nucleus.

4. This reviewer is not convinced that NADH specifically produced by solely the action of ALDH7A1 is the reason of the observed effect.

The kinetics with liposomes regarding Km values are impressive. However, no controls (negative) included in the experimental settings. That is just liposomes without any aldehyde (octanal of betaine aldehyde). It may be possible that some substrates of ALDH7A1 may be included in the liposomes that contribute to the catalytic activity of the enzyme.

5. Are there any other proteins besides ALDH7A1 interact with BARS? In addition to the limited information of the experiment performed, it is not clear if this interaction is observed in membranes or nucleus or cytosol.

6. The phosphorylation of ALDH7A1 by AMPK should be detected with mass spec to be more convincing.

7. It is a bit concerning that the authors have not discussed the already known functions of ALDH7A1 and how these correlate with their findings (if they do). These include the clinical phenotype of the ALDH7A1 mutations, the crystal structure of the protein as well as its role in osmosis and oxidative stress.

Overall, it is not really convincing, at least as presented, that the NADH generated solely by ALDH7A1 is the only reason for the observed changes.

Reviewer #2 (Remarks to the Author):

This study comprises different parts. The first part concludes that ATQ binds and inhibits the

fission protein BARS and therefore inhibits the intracellular trafficking pathways that depend on BARS. Moreover, it presents data that indirectly indicate that the BARS trafficking pathways are more numerous than those previously identified, and that BARS controls the majority of the intracellular traffic pathways (6 out of 8) of those examined in this study.

The second part concludes that the inhibition of traffic by ATQ is necessary to protect the cell from ATP depletion and cell death under hypoxia or starvation conditions.

The third part concludes that ATQ is a mediator of the protective action exerted by AMPK during hypoxia or starvation and analyzes the molecular mechanism of the interaction between AMPK and ATQ.

Finally, the last part analyzes the mechanism by which ATQ interacts with cell membranes and with BARS.

Main critical comments

This is a very complex and ambitious work that presents important conclusions. Some of these conclusions need to be strengthened with new experiments and/or discussed and justified with literature-based arguments in the revised text

One disconcerting aspect of the study is that it suggests that ATQ and AMPK exert their protective effects on cell survival and ATP content under conditions of hypoxia or starvation simply by inhibiting BARS-dependent intracellular trafficking. To our knowledge intracellular traffic has never been demonstrated or proposed to consume a large fraction of the cellular energy budget. On the contrary, it is thought that in eukaryotic cells most of the energy consumption is due to synthesis of proteins, lipids and other major cellular components. Furthermore, BARS-dependent pathways represent a fraction of the intracellular traffic system.

The authors should consider the possibility that BARS acts on cell functions other than traffic. In this regard, it should be noted that the BARS mutant insensitive to ATQ seems to completely abolish the effect of ATQ on trafficking, but only partially abolishes the protective effect of ATQ on cell death and energy depletion.

Another possibility that the authors should consider is that ATQ acts not only on BARS but also on other targets, possibly dehydrogenases, with different cellular functions.

The data are clear and must be accepted and presented, but they must be discussed on the basis of existing literature.

Along this line, it is known that AMPK modifies numerous substrates and acts on numerous cellular functions. The authors seem to propose that the protective effects of AMPK are mediated only by ATQ, but this does not seem to agree with the literature on the effects and mechanisms of action of AMPK. For example, the authors should examine the effect of AMPK also on transport routes that are not BARS and ATQ dependent. Or test whether a traffic-unrelated target of AMPK is affected by ATQ.

Minor comments

Was the interaction between ATQ and BARS already known? The authors do not describe how they arrive at this result and do not mention previous work. ... we identified antiquitin (ATQ) by mass spectrometry as a novel interacting protein of BARS. This experiment was done in this or in a previous study?

The transport routes known to be dependent on BARS are the retrograde KDEL transport, traffic from TGN to the plasma membrane and fluid phase endocytosis. What is the evidence that the other pathways examined in this study are BARS-dependent?

Not only NADH but also NAD can inhibit BARS-dependent fission. They use 10 micromolar NAD in the liposome experiment. Is this concentration inactive? They should show the data and discuss this point.

Fig. 5A is disorganized and confusing.

Reviewed by Alberto Luini

Reviewer #3 (Remarks to the Author):

This manuscript is centered on the mechanism through which the catalytic activity of Antiquitin (ATQ) regulates several intracellular transport pathways. It is proposed that ATQ, once phosphorylated by AMPK, interacts with the fission protein BARS, inhibiting its function, and thus membrane transport. This is a novel concept in that ATQ functions were so far related to its aldehyde dehydrogenase activity.

From a physiological point of view, the authors suggest that during hypoxia and starvation, the phosphorylated ATQ produces NADH that binds and inhibits BARS-mediated transport pathways. This decreased transport should be the cause of the decreased energy consumption during hypoxia and starvation.

The manuscript has several novel aspects that are of interest to the field. Some conclusions have however to be better substantiated as indicated below.

Major points:

A surprising conclusion in this manuscript refers to the fact that the energy consumption is mostly due to membrane transport. This sounds as a new concept not supported by the current literature on energy production and consumption involved in several cell processes such as protein synthesis and transport. This aspect should be discussed more thoroughly.

A novel information is the involvement of BARS in traffic steps (e.g., endocytic recycling of transferrin from RE to PM, cholera toxin internalization, EGFR transport from endosome to lysosome) other than those reported so far. This is an interesting aspect of the paper which needs however to be fully proven by indicating that these traffic steps cannot take place in the absence of BARS. The conclusion is now based on indirect data, i.e. ATQ SiRNA cells.

The ATQ-dependent regulation of BARS is confirmed on the basis of a negative control, the dehydrogenase ALDH1, that does not bind BARS and therefore does not affect membrane transport. This should be strengthened by additional controls such as the use of other dehydrogenases with different homology strength with BARS.

Concerning the mechanism of action of the AMPK-ATQ-BARS axis, according to the authors, AMPK phosphorylates and recruits ATQ onto the cellular membranes where it promotes NADH production. Is this enhanced NADH levels involved in the release of BARS from cellular membranes?

NADH favors the dimeric conformation of BARS: is the dimer released to the cytosol or is it bound to the membrane through ATQ? These potential mechanisms should be further analyzed or at least discussed with reference to the relevant literature.

The cell localization and activity of ATQ during starvation and hypoxia were not analyzed. The data reported are correlative and do not take into account, or at least discuss, other possible mechanisms that may involve ATQ in these processes.

Fig 2: The inhibition of BARS activity is usually associated with a fission-defective phenotype showing formation of tubules. ATQ depletion should induce a similar phenotype. Is this the case?

The expected ATQ SiRNA results in an enhancement of membrane transport mediated by BARS. This should not be the case if also BARS is KD: is this the case?

Fig 5F-G: The effect of BARS wt expression on ATP level and cell death should be analyzed and compared with the G172E mutant shown in the panels F-G.

The discussion should cover the comments made above and include the reference to the relevant literature.

Minor points:

Fig. 1: The MS methods and data should be specified in details unless published elsewhere. In the last case a reference should be provided.

Figs. 5J-K: these panels do not appear in the text. Either comment or delete.

In all Figure legends "n=10 fields of cells examined" is indicated. This should be better specified indicating for each experiment how many cells from how many independent experiments were analyzed.

RESPONSES TO REVIEWERS' COMMENTS

We wish to thank all reviewers for their thoughtful comments. In response, we have performed many new experiments. Because the incorporation of these new results required many previous figures to be re-numbered, we have highlighted in bold, both in the revised text and also in our responses below, when referring to new figures, so that the new data are more readily identified. In other cases, comments suggest confusion in the interpretation of our previous data. Thus, we have sought to address these comments through clarifications, as detailed in our responses below.

We would also like to mention two general changes. First, because the formal nomenclature for antiqutin is ALDH7A1, we have adopted ALDH7A1 in the revision. Second, as instructed by the new guidelines of the journal, we now show all quantitative data as individual data points for $n < 10$.

Overall, we appreciate that the collective revisions have further enhanced the presentation of our work.

Reviewer #1

General comment:

This is an interesting study and a number of experiments support in general their conclusion. However, all data are in one transformed cell line and there is a lack of a proposed mechanism regarding the substrates that are metabolized by ALDH7A1 and how they can contribute to the observed phenotype.

Regarding the first issue mentioned in the comment, we have now performed an extensive set of additional studies to show that the key findings are reproduced in another cell type, HEK293 cells (**new Figs S4A to S4M**).

Regarding the second issue, we wish to clarify that, rather than the metabolic products of ALDH7A1 activity, we have found that NADH generated by this activity underlies how ALDH7A1 inhibits the transport pathways, which is the phenotype being studied in the current manuscript.

The initial insight came from studies using the COPI vesicle reconstitution system, in which we examined a number of model metabolic products of ALDH7A1 and found that none could inhibit COPI vesicle formation (Fig 1H). Instead, led by the finding that ALDH7A1 inhibits COPI vesicle fission (Fig 1G), and also that NADH inhibits the fission role of BARS (Yang et al., 2005), we pursued the intriguing possibility that NADH generated by ALDH7A1 targets BARS to inhibit COPI vesicle fission. Initially, we found that simply adding NADH to the reconstitution system inhibited COPI vesicle formation (Fig 1H). We then performed a decisive experiment. A point mutation in BARS (G172E) that prevents NADH binding has been identified (Nardini et al., 2003). We replaced the wild-type BARS with this mutant BARS and

found that ALDH7A1 could no longer inhibit COPI transport (Figs 1I and 1J). Thus, with respect to the phenotype that we are studying, which is transport inhibition by ALDH7A1, the role of the metabolic conversion of ALDH7A1 activity is to support the generation of NADH, as the conversion of NAD to NADH by this activity requires the metabolic conversion of aldehydes to carboxylates.

We further note that precedence exists for the generation of NADH by the dehydrogenase activity of metabolic enzymes having primary cellular roles, rather than simply in supporting the catalytic activity of these enzymes. A well-known example involves the mitochondrial TCA cycle in which the NADH generated by different metabolic enzymes is utilized by the electron transport chain in generating a proton gradient to drive mitochondrial ATP production.

We also readily acknowledge that the metabolic activity of ALDH7A1 would have other roles besides simply supporting the generation of NADH for transport inhibition. In particular, we note that the focus of the current study is ALDH7A1 activity on membranes, because we have found that ALDH7A1 needs to be recruited to intracellular membrane compartments in order to exert transport inhibition. Thus, as ALDH7A1 has been suggested in previous studies to be protective against lipid peroxidation (Brocker et al., 2011; Brocker et al., 2010), which is a form of oxidative stress that also occurs on membranes, an intriguing possibility is that ALDH7A1 activity on membranes coordinates two general processes that protect against two major types of cellular stress: i) inducing transport inhibition to reduce energy consumption in protecting against energy stress, which is mediated through the generation of NADH, and ii) reducing lipid peroxidation in protecting against oxidative stress, which is mediated through the metabolic conversion of aldehydes to carboxylates. We have now added this discussion in providing a more encompassing view of how ALDH7A1 activity on membranes could be acting in the cell. This is highlighted in italics in a revised Discussion section.

1. The approved name of the gene/protein (HGNC) is ALDH7A1 and this should be used throughout the manuscript. The use of the alias ALDH7A1 creates confusion. The same thing for ALDH1. Is this ALDH1A1 as it is noted in one for the figures? ALDH1 subfamily consists of many genes including ALDH1A1, 1A2, 1A3, 1B1, 1L1 and 1L2. To avoid confusion in the literature and to be consistent with the journals policy, HGNC approved symbols should be used.

We have now revised the text to replace antiquitin with ALDH7A1. We have also revised the presentation of the previous data that examined ALDH1 to clarify that ALDH1A1 was examined. This data was previously shown as Figure S3 and now shown as Figure S6.

2. The authors claim that the NADH produced by ALDH7A1 is the major player for the function of this gene product. However there is not any mentioning about the substrates metabolized by this protein, as well as their importance.

We have sought to clarify above (see our response above to the general comment by reviewer) the key experimental evidence that has led us to conclude that NADH generated by ALDH7A1 activity, rather than the carboxylates generated by this activity, explains how

transport inhibition occurs. Thus, the role of the metabolic conversion with respect to the phenotype that we are studying, which is transport inhibition, is to support NADH generation. Notably, we have also discussed another cellular role for the metabolic conversion by ALDH7A1 activity. Led by the consideration that we are studying ALDH7A1 activity on membranes, and that ALDH7A1 has been suggested to reduce lipid peroxidation, a process that also occurs on membranes, we proposed the intriguing possibility is that ALDH7A1 on membranes coordinates two general processes that mitigate against two major types of cellular stress: i) inhibiting transport to reduce cellular energy consumption in reducing energy stress, and ii) reducing lipid peroxidation in reducing oxidative stress.

3. It is a little bit confusing when it comes to the cellular localization of the effects. Are they membrane specific? do they occur in the cytosol? Is this phenotype related to the nucleus? Brocker et al, have shown that this ALDH7A1 gene is expressed in mitochondria, cytosol and nucleus.

In the original submission, we provided multiple lines of evidence to support that ALDH7A1 needs to be recruited from the cytosol to cytoplasmic membranes to exert transport inhibition. First, we have found that BARS only interacts with ALDH7A1 on cytoplasmic membranes (previously Fig 7G, now Fig S7A). Second, we found that the replacement of endogenous ALDH7A1 with the S102D mutant, which renders ALDH7A1 largely bound to cytoplasmic membranes (see Figs 7B and 7C), is sufficient to exert transport inhibition in the normal (non-hypoxic) condition (see Figs 6G and 6H). Third, we found that the replacement of endogenous ALDH7A1 with the S102A mutant, which renders ALDH7A1 largely soluble (see Figs 7B and 7C), prevents its ability to mediate transport inhibition induced by hypoxia (see Figs 6E and 6F).

To further support our previous findings, we have now performed additional studies. We find that starvation and hypoxia, which are conditions that activate AMPK to phosphorylate ALDH7A1, induce cytosolic ALDH7A1 to be recruited to cytoplasmic membranes (**new Fig S10C**). Further characterizing these membranes, we have performed confocal microscopy, which reveals that ALDH7A1 is recruited to multiple cytoplasmic compartments involved in vesicular transport (**new Fig S10D**). Notably, ALDH7A1 does not show enhanced recruitment to the endoplasmic reticulum (ER), which is consistent with our finding that ALDH7A1 does not affect transport from the ER.

4. This reviewer is not convinced that NADH specifically produced by solely the action of ALDH7A1 is the reason of the observed effect. The kinetics with liposomes regarding Km values are impressive. However, no controls (negative) included in the experimental settings. That is just liposomes without any aldehyde (octanal or betaine aldehyde). It may be possible that some substrates of ALDH7A1 may be included in the liposomes that contribute to the catalytic activity of the enzyme.

We have now performed additional controls. When no substrate is added to the reconstitution of ALDH7A1 activity using liposomes, we find that no ALDH7A1 activity occurs (**new Fig S11A**). Moreover, when no NAD is added to this reconstitution system, no ALDH7A1 activity occurs (**new Fig S11A**).

5. Are there any other proteins besides ALDH7A1 interact with BARS? In addition to the limited information of the experiment performed, it is not clear if this interaction is observed in membranes or nucleus or cytosol.

To address the first part of the comment, we wish to clarify that we had identified multiple interacting proteins of BARS. This involved the incubation of GST-BARS with cytosol followed by mass spectrometry to identify proteins that interact with BARS. In light of the comment, we now list all interacting proteins (**new Fig S1A**). We were intrigued by the metabolic enzymes on this list (highlighted in red in **Fig S1A**), because of our recent finding that GAPDH exerts transport inhibition by targeting ARF GAPs (Yang et al., 2018). Thus, pursuing further studies on these metabolic enzymes, we identified ALDH7A1 as another metabolic enzyme that inhibits COPI transport (**new Figs S1B to S1E**).

Regarding the second part of the comment, we wish to clarify that we had shown in the original submission that ALDH7A1 interacts with BARS on cytoplasmic membranes and not in the cytosol (previously Fig 7G, now Fig S7A).

6. The phosphorylation of ALDH7A1 by AMPK should be detected with mass spec to be more convincing.

As suggested by the comment, we initially attempted mass spectrometry (MS). However, our core facility was unsuccessful in detecting the phosphorylation of the S102 residue in ALDH7A1 by AMPK. Because this was a negative result, and also because we were told that, depending on the protein, MS detection of phosphorylation can be challenging, we then pursued another approach. Prior to the advent of MS technology, the standard way of showing that a kinase directly phosphorylates a substrate involves the radiolabeling of the substrate with P32 derived from ATP. Notably, with this approach, we found by the kinase assay that the incubation of AMPK and ALDH7A1, both as recombinant proteins, resulted in the P32 labeling of the wild-type, but not the S102A mutant, of ALDH7A1 (**new Fig S10A**).

We then sought another way of confirming the above result. Because kinases phosphorylate their substrates by transferring the gamma-phosphate on ATP to substrates, kinase activity can be monitored by quantifying the level of ADP generated. An additional advantage of this approach is that it also provides insight into the efficiency by which a kinase phosphorylates a substrate. We had recently pursued this quantitative approach to show that AMPK phosphorylates GAPDH with reasonable efficiency, suggesting that this phosphorylation has physiologic relevance (Yang et al., 2018). Revisiting this approach, we have as not only confirmed that AMPK phosphorylates ALDH7A1 at its S102 residue, as only the incubation of the wild-type ALDH7A1, but not the S102A mutant, with AMPK resulted in the generation of ADP (**new Fig S10B**), but also obtained the stoichiometry of this phosphorylation (**new Fig**

S10B), which reveals that it is similar to what we had observed previously for AMPK phosphorylating GAPDH (Yang et al., 2018). To facilitate this comparison for the reviewer, we have included both analyses, our previous result on AMPK phosphorylating GAPDH and our current result on AMPK phosphorylating ALDH7A1, as a figure at the end of the response document. Note that SAMS is a peptide derived from an optimal substrate of AMPK, acetyl-coA carboxylase (ACC), and thus, has been used widely as a standard when comparing AMPK phosphorylation of other substrates.

Overall, we have taken three complementary approaches to show that AMPK phosphorylates ALDH7A1 at its S102 residue. In the original submission, we used an antibody-based approach. In this revision, we have added a radiolabeling approach, and also a quantitative approach that enables us to assess the efficiency by which a kinase phosphorylates a substrate.

7. It is a bit concerning that the authors have not discussed the already known functions of ALDH7A1 and how these correlate with their findings (if they do). These include the clinical phenotype of the ALDH7A1 mutations, the crystal structure of the protein as well as its role in osmosis and oxidative stress.

We note that the clinical phenotype of ALDH7A1 mutations involves its role in lysine metabolism, which involves ALDH7A1 acting on a soluble substrate (Mills et al., 2006). In contrast, transport inhibition involves ALDH7A1 acting on the membrane. Moreover, we have discussed above (see our responses to comments #1 and #2) how membrane-bound ALDH7A1 activity could be coordinating two processes to protect against two types of cellular stress: i) inducing transport inhibition to reduce energy consumption in protecting against energy stress, which is mediated through the generation of NADH, and ii) reducing lipid peroxidation to protect against oxidative stress, which is mediated through the metabolic conversion of aldehydes to carboxylates.

Reviewer #2

Main critical comments

1. This is a very complex and ambitious work that presents important conclusions. Some of these conclusions need to be strengthened with new experiments and/or discussed and justified with literature-based arguments in the revised text.

We thank the reviewer for lauding the overall significance of our work. We have added new experiments and revised the text in response to the multiple comments provided by the reviewer, as detailed below.

2. One disconcerting aspect of the study is that it suggests that ALDH7A1 and AMPK exert their

protective effects on cell survival and ATP content under conditions of hypoxia or starvation simply by inhibiting BARS-dependent intracellular trafficking. To our knowledge intracellular traffic has never been demonstrated or proposed to consume a large fraction of the cellular energy budget. On the contrary, it is thought that in eukaryotic cells most of the energy consumption is due to synthesis of proteins, lipids and other major cellular components. Furthermore, BARS-dependent pathways represent a fraction of the intracellular traffic system.

Regarding the first part of the comment, we completely agree that the intracellular transport pathways had not been known to be a major target by which cellular energy consumption can be reduced for energy homeostasis. As explanation, we note that vesicular transport has been typically studied with respect to single pathways. In contrast, we have identified ALDH7A1 to inhibit multiple transport pathways. By imposing quantitative inhibition across many pathways, our results reveal that a meaningful reduction in cellular energy consumption can be achieved in promoting energy homeostasis during hypoxia and starvation. Further supporting this explanation, we have recently identified another metabolic enzyme, GAPDH, also to exert a broad inhibition of the transport pathways in promoting energy homeostasis, in this case during starvation (Yang et al., 2018). However, whereas GAPDH targets ARF GAPs for transport inhibition, ALDH7A1 targets a different class of transport factors, for which we have identified BARS as one such factor. Overall, the results from the two studies now point to the importance of targeting the transport pathways globally, rather than individually, as a way of achieving a significant reduction in energy consumption for homeostasis during situations of energy deprivation. We thank the reviewer for pointing out the need for such a discussion and have added it to a revised Discussion, which is highlighted in italics.

Regarding the second part of the comment, that BARS-dependent pathways represent a fraction of the intracellular traffic system, see our response to the comment below (comment #3), as the issue has been re-iterated in further detail.

3. The authors should consider the possibility that BARS acts on cell functions other than traffic. In this regard, it should be noted that the BARS mutant insensitive to ALDH7A1 seems to completely abolish the effect of ALDH7A1 on trafficking, but only partially abolishes the protective effect of ALDH7A1 on cell death and energy depletion. Another possibility that the authors should consider is that ALDH7A1 acts not only on BARS but also on other targets, possibly dehydrogenases, with different cellular functions.

BARS is currently known to operate in three of the six major pathways that we have found to be targeted by ALDH7A1 for inhibition. These are Golgi to ER transport, Golgi to plasma membrane transport, and fluid-phase endocytosis (Bonazzi et al., 2005; Yang et al., 2005). To address the comment, we have now performed siRNA against BARS to show that only these currently known pathways are regulated by BARS. In contrast, the other three pathways that we have found to be targeted by ALDH7A1 are not affected by siRNA against BARS, and thus are BARS-independent pathways (**new Figs S5A to S5F**). Notably, this finding also suggests why inhibition through ALDH7A1 promotes energy homeostasis more

effectively than inhibition through BARS. Whereas ALDH7A1 controls six of the eight major intracellular pathways that we have examined, BARS controls only three of these pathways.

With respect to other possibilities mentioned by the comment, we would like to point out that we have already examined the other known role of BARS, which acts in transcription repression (known as CtBP in this role), and find that ALDH7A1 does not inhibit transport through this other function of BARS/CtBP (previously Fig S4E to S4I, now Fig S7D to S7H). We also readily acknowledge that ALDH7A1 could have other cellular effects other than transport inhibition. However, because this study deals with how ALDH7A1 exerts transport inhibition in promoting energy homeostasis, we hope the reviewer can understand why we believe that the exploration of other cellular effects of ALDH7A1 should be considered a future goal.

4. The data are clear and must be accepted and presented, but they must be discussed on the basis of existing literature. Along this line, it is known that AMPK modifies numerous substrates and acts on numerous cellular functions. The authors seem to propose that the protective effects of AMPK are mediated only by ALDH7A1, but this does not seem to agree with the literature on the effects and mechanisms of action of AMPK. For example, the authors should examine the effect of AMPK also on transport routes that are not BARS and ALDH7A1 dependent. Or test whether a traffic-unrelated target of AMPK is affected by ALDH7A1.

We apologize for having given the mis-impression that ALDH7A1 mediates all the effects of AMPK on energy homeostasis. This is clearly not the case, as AMPK is known to affect a variety of cellular processes for its role as the master coordinator of cellular energy homeostasis (Hardie et al., 2012). We have also performed specific experiments requested by the comment. First, we find that clathrin endocytosis, which is not regulated by either ALDH7A1 (Fig 2G) or BARS (Bonazzi et al., 2005), is not regulated by AMPK (**new Fig S9H**). Second, as AMPK has been shown to activate sirtuin 1 (Sirt1) activity in modulating transcription (Chang et al., 2015), we find that siRNA against ALDH7A1 does not prevent AICAR (a pharmacologic activator of AMPK) from activating Sirt1 activity (**new Fig S9G**).

Minor comments

1. Was the interaction between ALDH7A1 and BARS already known? The authors do not describe how they arrive at this result and do not mention previous work. ... we identified antiquitin (ALDH7A1) by mass spectrometry as a novel interacting protein of BARS. This experiment was done in this or in a previous study?

We had identified multiple interacting proteins of BARS as part of this study. Thus, in response to the comment, we now show the other proteins (**new Fig S1A**). In light of our recent finding that GAPDH exerts a broad inhibition of the transport pathways by targeting ARF GAPs (Yang et al., 2018), we pursued further studies on metabolic enzymes found to associate with BARS (**new Figs S1B to S1E**), which then led us to identify ALDH7A1 as another metabolic

that inhibits the intracellular pathways through an unexpected mechanism.

2. The transport routes known to be dependent on BARS are the retrograde KDEL transport, traffic from TGN to the plasma membrane and fluid phase endocytosis. What is the evidence that the other pathways examined in this study are BARS-dependent?

See our response to major comment #3 above, in which we have addressed this question.

3. Not only NADH but also NAD can inhibit BARS-dependent fission. They use 10 micromolar NAD in the liposome experiment. Is this concentration inactive? They should show the data and discuss this point.

We wish to clarify that the 10 μ M of NAD used in the liposome experiment refers to our study that reconstitutes ALDH7A1 activity on membranes using liposomes. For this activity to occur, it requires substrate and NAD (for which we added 10 μ M). The other experiment that involves the use of liposomes is the BARS tubulation assay (previously S1D, now S2D), in which we only used NADH.

4. Fig. 5A is disorganized and confusing.

Fig 5A presents data similarly as those shown in Figs 5B, 5H, and 5I. Thus, it is unclear to us why Fig 5A has been singled out. We are happy to re-organize this figure, if the reviewer can tell us more specifically what needs to be re-organized to reduce confusion.

Reviewer #3

Major points:

1. A surprising conclusion in this manuscript refers to the fact that the energy consumption is mostly due to membrane transport. This sounds as a new concept not supported by the current literature on energy production and consumption involved in several cell processes such as protein synthesis and transport. This aspect should be discussed more thoroughly.

We completely agree that the intracellular transport pathways had not been known to be a major target by which cellular energy consumption can be reduced for energy homeostasis. As explanation, we note that studies on vesicular transport have typically focused on single pathways. In contrast, we have identified ALDH7A1 to exert a broad inhibition of the transport pathways. Notably, by imposing quantitative inhibition across many pathways, our results reveal that a meaningful reduction in cellular energy consumption can be achieved for energy homeostasis during hypoxia and starvation. Supporting this explanation, we have recently identified another factor, GAPDH, which can also exert a broad inhibition of the transport pathways, and by doing so also promote energy homeostasis during starvation (Yang et al., 2018). However, whereas GAPDH targets ARF GAPs for transport inhibition, ALDH7A1

targets a different class of transport factors, for which we have identified BARS as one such factor. Nevertheless, the collective results now point to the importance of targeting the transport pathways globally, rather than individually, as a way of achieving a significant reduction in energy consumption for homeostasis during situations of energy deprivation.

We thank the reviewer for pointing out the need for such a discussion and have added it to a revised Discussion, which is highlighted in italics.

2. A novel information is the involvement of BARS in traffic steps (e.g., endocytic recycling of transferrin from RE to PM, cholera toxin internalization, EGFR transport from endosome to lysosome) other than those reported so far. This is an interesting aspect of the paper which needs however to be fully proven by indicating that these traffic steps cannot take place in the absence of BARS. The conclusion is now based on indirect data, i.e. ALDH7A1 siRNA cells.

BARS is known to operate in only three of the six pathways that we have found to be targeted by ALDH7A1, which are Golgi to ER transport, Golgi to plasma membrane transport, and fluid-phase endocytosis (Bonazzi et al., 2005; Yang et al., 2005). As confirmation, we have now performed siRNA against BARS to show that only these pathways are inhibited by the treatment. In contrast, three other pathways targeted by ALDH7A1 are not affected by this treatment, and thus are BARS-independent pathways (**new Figs S5A to S5F**). Notably, this finding explains why inhibition of ALDH7A1 affects energy homeostasis more strongly than the use of the BARS mutant that is insensitive to NADH (compare Figs 5A and 5B with Figs 5H and 5I).

We would also like to suggest that the identification of target(s) by which ALDH7A1 inhibits the BARS-independent pathways should be considered a future goal, as none of the currently known transport factors operating in the BARS-independent pathways targeted by ALDH7A1 (which are endocytic recycling, endocytic transport to the Golgi, and endocytic transport to the lysosome) are known to be regulated by NADH binding. Thus, we would need to embark on essentially an open-ended search. More importantly, the ultimate identification of such target(s) would not affect the main point of the current study, which is that we have identified a novel role of ALDH7A1, exerting a broad inhibition of the transport pathways to reduce energy consumption during starvation and hypoxia. Thus, we are asking for a reasonable assessment of what can be accomplished in a single study.

3. The ALDH7A1-dependent regulation of BARS is confirmed on the basis of a negative control, the dehydrogenase ALDH1, that does not bind BARS and therefore does not affect membrane transport. This should be strengthened by additional controls such as the use of other dehydrogenases with different homology strength with BARS.

We have now examined two other metabolic enzymes with dehydrogenase activity that are not aldehyde dehydrogenases, aldolase and malate dehydrogenase. We find that siRNA against either does not affect fluid-phase endocytosis (**new Figs S5G to S5I**), which is a BARS-dependent pathway that we had found to be targeted by ALDH7A1.

4. Concerning the mechanism of action of the AMPK-ALDH7A1-BARS axis, according to the authors, AMPK phosphorylates and recruits ALDH7A1 onto the cellular membranes where it promotes NADH production. Is this enhanced NADH levels involved in the release of BARS from cellular membranes?

We have performed the requested study, and find that BARS is indeed released from this membrane upon incubation with the S102D mutant of ALDH7A1 (**new Fig 7G**), which generates a high level of NADH on membranes (see Fig 7E). In contrast, incubation of Golgi membrane with the S102A mutant of ALDH7A1, which generates markedly less NADH (see Fig 7F), had markedly reduced capacity to induce the release of BARS (**new Fig 7G**).

5. NADH favors the dimeric conformation of BARS: is the dimer released to the cytosol or is it bound to the membrane through ALDH7A1? These potential mechanisms should be further analyzed or at least discussed with reference to the relevant literature.

We have performed the requested analysis. We examined BARS that had been released from Golgi membrane upon incubation with the S102D mutant of ALDH7A1 (**new Fig 7G**). Gel filtration confirms that the released BARS exists as a dimer (**new Fig S11B**).

We further note that BARS has been suggested to adopt an open conformation on membranes and a closed conformation in solution (Nardini et al., 2003). Thus, we propose the following sequence of events: AMPK phosphorylation of ALDH7A1 induces its recruitment to membranes to interact with BARS in its open conformation. This association on membranes also allows NADH generated by ALDH7A1 to be efficiently targeted to BARS. Upon NADH binding, BARS is converted from an open conformation to a closed conformation, which results in the dimerization of BARS and its release from membranes.

6. The cell localization and activity of ALDH7A1 during starvation and hypoxia were not analyzed. The data reported are correlative and do not take into account, or at least discuss, other possible mechanisms that may involve ALDH7A1 in these processes.

To address the first part of the comment, we have performed subcellular fractionation showing that starvation and hypoxia redistributes ALDH7A1 from the cytosol to membranes (**new Fig S10C**). We have also performed confocal microscopy to further characterize the membrane compartments to which ALDH7A1 is recruited (**new Fig S10D**).

To address the second part of the comment, we acknowledge that ALDH7A1 could have other roles in response to starvation and hypoxia. One possibility is suggested by previous studies that have found ALDH7A1 activity to ameliorate lipid peroxidation (Brocker et al., 2011; Brocker et al., 2010). This pathologic condition is a form of oxidative stress that also occurs on membranes (Reed, 2011). Thus, an intriguing possibility is that ALDH7A1 activity on membranes coordinates two overall processes: i) inducing transport inhibition to reduce energy consumption, which is mediated through the generation of NADH, and ii) ameliorating lipid

peroxidation to reduce oxidative stress, which is mediated through the metabolic conversion of aldehydes to carboxylates. We have added this discussion to the revised Discussion section, which is highlighted in italics.

7. Fig 2: The inhibition of BARS activity is usually associated with a fission-defective phenotype showing formation of tubules. ALDH7A1 depletion should induce a similar phenotype. Is this the case?

We are assuming that this comment is asking whether ALDH7A1 activation (rather than ALDH7A1 depletion as noted by the comment) would reproduce the effect of BARS inhibition, which is the mechanism that we have elucidated in this study. Thus, we have performed the following experiment. Inhibition of BARS has been shown previously to accumulate Golgi tubules, because these tubules require BARS-mediated fission for detachment from the Golgi (Bonazzi et al., 2005). Revisiting this assay, we find that the expression of the S102D mutant of ALDH7A1 also induces the accumulation of Golgi tubules (**new Figs 7H and S11C**).

8. The expected ALDH7A1 siRNA results in an enhancement of membrane transport mediated by BARS. This should not be the case if also BARS is KD: is this the case?

As requested, we have now confirmed that siRNA against ALDH7A1 can no longer enhance COPI transport, when cells are also treated with siRNA against BARS, which inhibits this pathway (**new Fig S3I**).

9. Fig 5F-G: The effect of BARS wt expression on ATP level and cell death should be analyzed and compared with the G172E mutant shown in the panels F-G.

We have now performed the requested experiment, which shows that the expression of wild-type BARS is similar to that of control (**revised Figs 5H and 5I**). Technically, we replaced endogenous BARS with transfected BARS, so that BARS is not overexpressed.

10. The discussion should cover the comments made above and include the reference to the relevant literature.

As requested, we have now revised the discussion to incorporate results from the above comments by this reviewer.

Minor points:

1. Fig. 1: The MS methods and data should be specified in details unless published elsewhere. In the last case a reference should be provided.

The MS data was generated in this study. Thus, in response to the comment, we now show the result (**new Fig S1A**). In light of our recent finding that GAPDH exerts a broad

inhibition of the transport pathways by targeting ARF GAPs (Yang et al., 2018), we pursued further studies on metabolic enzymes found to associate with BARS (**new Figs S1B to S1E**), which then led us to identify ALDH7A1 as another metabolic that inhibits the intracellular pathways through an unexpected mechanism.

2. *Figs. 5J-K: these panels do not appear in the text. Either comment or delete.*

We thank the reviewer for pointing out this omission, and have added text to describe these results, which are now re-numbered as Figure 5F and 5G.

3. *In all Figure legends “n=10 fields of cells examined” is indicated. This should be better specified indicating for each experiment how many cells from how many independent experiments were analyzed.*

We wish to clarify that each field has about 3 cells. However, because this number can vary from field to field, but the journal guideline requires that we provide an exact number, we indicated 10 fields of cells examined. Moreover, as requested, we have specified in the figure legends the number of independent experiments performed.

REFERENCES

Bonazzi, M., Spano, S., Turacchio, G., Cericola, C., Valente, C., Colanzi, A., Kweon, H.S., Hsu, V.W., Polishchuck, E.V., Polishchuck, R.S., *et al.* (2005). CtBP3/BARS drives membrane fission in dynamin-independent transport pathways. *Nat Cell Biol* 7, 570-580.

Brocker, C., Cantore, M., Failli, P., and Vasiliou, V. (2011). Aldehyde dehydrogenase 7A1 (ALDH7A1) attenuates reactive aldehyde and oxidative stress induced cytotoxicity. *Chem Biol Interact* 191, 269-277.

Brocker, C., Lassen, N., Estey, T., Pappa, A., Cantore, M., Orlova, V.V., Chavakis, T., Kavanagh, K.L., Oppermann, U., and Vasiliou, V. (2010). Aldehyde dehydrogenase 7A1 (ALDH7A1) is a novel enzyme involved in cellular defense against hyperosmotic stress. *J Biol Chem* 285, 18452-18463.

Chang, C., Su, H., Zhang, D., Wang, Y., Shen, Q., Liu, B., Huang, R., Zhou, T., Peng, C., Wong, C.C., *et al.* (2015). AMPK-Dependent Phosphorylation of GAPDH Triggers Sirt1 Activation and Is Necessary for Autophagy upon Glucose Starvation. *Mol Cell* 60, 930-940.

Hardie, D.G., Ross, F.A., and Hawley, S.A. (2012). AMPK: a nutrient and energy sensor that maintains energy homeostasis. *Nat Rev Mol Cell Biol* 13, 251-262.

Mills, P.B., Struys, E., Jakobs, C., Plecko, B., Baxter, P., Baumgartner, M., Willemsen, M.A., Omran, H., Tacke, U., Uhlenberg, B., *et al.* (2006). Mutations in antiquitin in individuals with pyridoxine-dependent seizures. *Nat Med* 12, 307-309.

Nardini, M., Spano, S., Cericola, C., Pesce, A., Massaro, A., Millo, E., Luini, A., Corda, D., and Bolognesi, M. (2003). CtBP/BARS: a dual-function protein involved in transcription co-repression and Golgi membrane fission. *Embo J* 22, 3122-3130.

Reed, T.T. (2011). Lipid peroxidation and neurodegenerative disease. *Free radical biology & medicine* 51, 1302-1319.

Yang, J.S., Hsu, J.W., Park, S.Y., Li, J., Oldham, W.M., Beznoussenko, G.V., Mironov, A.A., Loscalzo, J., and Hsu, V.W. (2018). GAPDH inhibits intracellular pathways during starvation for cellular energy homeostasis. *Nature* 561, 263-267.

Yang, J.S., Lee, S.Y., Spanò, S., Gad, H., Zhang, L., Nie, Z., Bonazzi, M., Corda, D., Luini, A., and Hsu, V.W. (2005). A role for BARS at the fission step of COPI vesicle formation from Golgi membrane. *EMBO J* 24, 4133-4143.

Phosphorylation stoichiometry

AMPK phosphorylating GAPDH
(Nature 2018, 561: 263-7)

AMPK phosphorylating ALDH7A1
(Current study)

	Initial rate of phosphorylation (pmol / min)	Stoichiometry of maximal phosphorylation (mol substrate / mol kinase)	Stoichiometry of maximal phosphorylation (mol phosphorylation / mol total substrate)
SAMS	147±13	2120±321	0.816±0.124
GAPDH	31.5±13	1022±57	0.662±0.037

	Initial rate of phosphorylation (pmol / min)	Stoichiometry of maximal phosphorylation (mol substrate / mol kinase)	Stoichiometry of maximal phosphorylation (mol phosphorylation / mol total substrate)
SAMS	114.9 ± 17.5	2081 ± 209	0.801 ± 0.080
ALDH7A1	29.8 ± 8.5	808 ± 131	0.566 ± 0.092

REVIEWERS' COMMENTS:

Reviewer #1 (Remarks to the Author):

The authors has addressed most of my concerns.

Reviewer #2 (Remarks to the Author):

The authors have addressed my concerns and requests in full in the revised version of the manuscript.

In my view this is an ambitious and interesting study that elucidates important regulatory components of the BARS-dependent fission machinery and unravels previously unsuspected links between this machinery and the processes of energy production and cell survival.

Reviewer #3 (Remarks to the Author):

The paper by V. Hsu and co-workers entitled "ALDH7A1 exerts a broad inhibition of the intracellular transport pathways during hypoxia and starvation to promote cellular energy homeostasis" now includes new data and comments that address all points I have raised in my first review.

In particular, and due to the new experimental part, the author' order has been changed, the title now is aligned with the reported results, new figures in the main text and as supplementary material have been provided.

Altogether I find the present manuscript of good quality and in line with the quality requirements of Nature Communications.